# EIGENLORA: RECYCLE TRAINED ADAPTERS FOR RESOURCE EFFICIENT ADAPTATION AND INFERENCE

## ABSTRACT

Low-Rank Adapters (LoRA) are lightweight components that have made fine-tuning large models on domain-specific tasks inexpensive. This has resulted in an abundance of adapters in a growing open-source public community. We ask the question: can these adapters be used to inform and further streamline adaptation to new tasks? We introduce EigenLoRA, a parameter-efficient fine-tuning method that uses trained adapters to perform fast adaptation on new domains with orders of magnitude fewer parameters than LoRA. Our method finds a principal subspace that aligns with the domain of the trained adapters. This allows for efficient and fast adaptation to new tasks in this domain by simply learning coefficients on the principal components of this subspace. Furthermore, EigenLoRA makes inference time task-switching memory efficient. Instead of saving and loading whole LoRAs, EigenLoRA can simply load lightweight coefficients. EigenLoRA[1] works across a variety of domains and tasks and is a viable solution for edge-based and efficient personalization applications.

## 1 INTRODUCTION

Rapid progress in the fields of natural language processing (Touvron et al., 2023) and computer vision (Rombach et al., 2021) has fueled the development of ever-growing large-scale models where training billions of parameters has become commonplace. The size and complexity of these models have made it infeasible for the average researcher to train or finetune them on downstream tasks and datasets. In order to overcome these challenges, there has been an increased interest in parameter-efficient fine-tuning (PEFT) methods like adapters (Houlsby et al., 2019; Chen et al., 2022; Luo et al., 2023), low rank adaptation (LoRA) methods (Hu et al., 2021; Kopiczko et al., 2023; Liu et al., 2024), prompt-based methods (Lester et al., 2021; Razdaibiedina et al., 2023; Fischer et al., 2024), etc.

In particular, LoRA and related follow-up works (Meng et al., 2024; Liu et al., 2024) have garnered significant interest due to their simplicity and effectiveness. This simplicity of usage has led to a proliferation of low-rank adapters within an expanding open-source community. These adapters contain compressed information from their training data, which may or may not be publicly available, inspiring an examination of whether adapter weights can be used to inform and enhance the efficiency of adaptation to new tasks. Recent work has shown that weight updates to deep neural networks occurs in low-dimensional invariant subspaces (Kwon et al., 2024). This raises a possible hypothesis that these LoRA adapters may also share such a *principal subspace* that could be reused without having to search for it from scratch during the training of new adapters. We introduce **EigenLoRA**, a PEFT method that extracts a set of information-dense principal components defining a subspace, by decomposing the weights of a given set of trained adapters. This allows us to reduce the number of learnable parameters (extending up to $100\times$ less than LoRA) and achieve faster optimization (upto $2\times$) of new adapters. Moreover, EigenLoRA allows for more memory-efficient inference using multiple task adapters, especially benefiting edge devices (Liu et al., 2022). We also present a theoretical analysis of our method providing an approximation bound on reconstruction error when projecting to principal subspaces; and our experiments across a wide range of vision and language related tasks demonstrate its wide applicability.

Figure 1 shows an overview of how our method works. In summary, we propose **EigenLoRA** – a method to recycle trained adapters by identifying a *task-invariant* weight subspace that is shared by

---

[1]We will release code compatible with HuggingFace PEFT and Diffusers library for EigenLoRA.

Figure 1: LoRA uses low rank matrices ($r \ll n$) for task adaptation. We observe that domain-specific tasks may share a subspace even in the smaller LoRA weight space. This allows us to extract *task-invariant* principal components defining this subspace. Using these fixed components, each LoRA can be represented using an even smaller number of *task-specific* coefficients ($K \ll n$), making adapter training fast, and more parameter efficient; and inference more memory efficient.

all tasks in the adapter domain. We hypothesize (and validate through experiments) that weights for different tasks in a domain are restricted to this subspace. This restriction allows for more efficient training of new adapters as they can be learned faster with fewer *task-specific* parameters; and multiple adapters can be served with lower memory footprint, improving inference efficiency. Our key contributions are as follows:

- **(Training)**: EigenLoRA uses significantly **fewer number of parameters** (up to $100\times$) to train than LoRA, and **converges faster** (up to $2\times$) than comparable methods, while reaching similar or better performance.
- **(Inference)**: EigenLoRA improves the **memory efficiency of inference** ($\approx 18\times$) on multiple tasks, by reducing the number of switchable parameters between tasks.
- **(Applicability)**: We empirically demonstrate the effectiveness of EigenLoRA on a range of aligned and diverse domains across different modalities of data (text/image). This validates the existence of shared principal subspaces in modalities across the board.

## 2 RELATED WORKS

Low-Rank Adaptation refers to modeling neural network weight updates as a function of low-rank matrices instead of training the entire weight matrix. This is a well-established line of research starting from Burer-Monteiro factorization (Burer & Monteiro, 2003), with a recent resurgence by Hu et al. (2021) (LoRA), who used it as a technique to finetune LLMs; and other related variants (Ma et al., 2024; Chi et al., 2019; Kwon et al., 2024). However, with rapid growth in the scale of models, Low-Rank Adaptation has also become relatively expensive; for example, LoRA with a rank of 16 on GPT-3 Brown et al. (2020) requires 75.5 million parameters. Consequently, more efficient low-rank fine-tuning methods are being developed. Mixture of experts models (Huang et al., 2023; Wu et al., 2024; Diao et al., 2023; Zhong et al., 2024; Zhou et al., 2018) have been proposed as a method to adapt to new domains using a mixture of low-rank modules. But these approaches typically require a substantial number of high-quality adapters to work efficiently (Ku et al., 2024), which can significantly increase the model memory requirements (Zhou et al., 2022). Furthermore, complex gating or weighting mechanisms utilized with these models can exhibit training instability (Zoph et al., 2022).

Recent methods have aimed to learn better subspaces for low-rank optimization, primarily by decomposing model weights into singular vectors for improved training. Meng et al. (2024) demonstrate that initializing LoRA with singular vectors is superior to random initialization, while Sharma et al. (2023) find that removing minor singular components enhances robustness. Using randomly initialized principal components (Kopiczko et al., 2023) or weight matrices (Koohpayegani et al., 2024) has also been explored to reduce the number of trainable parameters. However, as shown

in Section 4, random initialized subspaces may not be very useful. This is intuitive as the random subspace may not have an overlap with domain-specific principal subspaces. On the other hand, EigenLoRA uses trained adapters to extract a *principal subspace* suitable for a given domain of tasks resulting in a better subspace initialization than and parameter efficiency. Given our focus on resource and computation efficiency in this work, we focus primarily on LoRA (Hu et al., 2021) as our main baseline, but EigenLoRA can be used with any PEFT method like Liu et al. (2024); Zhang et al. (2023) where task-specific weights can be analyzed together.

## 3 METHOD

In this section, we describe the theoretical motivation and the algorithm of our method, with a discussion on the hyper-parameters and quantification of practical benefits.

### 3.1 THEORETICAL MOTIVATION

Let $W \in \mathbb{R}^{m \times n}$ be a linear transformation matrix from vector space $\mathbb{R}^m$ to $\mathbb{R}^n$. If $W$ is a full-rank (with rank $min(n, m)$) transformation matrix, then it represents all possible linear mappings between the two spaces. In contrast, LoRA adapters are defined as two matrices $B \in \mathbb{R}^{m \times r}$ and $A \in \mathbb{R}^{r \times n}$ such that $BA$ has the same size as $W$ but rank $r < min(n, m)$. These matrices combine to yield a linear transformation between the same spaces $\mathbb{R}^m$ to $\mathbb{R}^n$, but cannot span the entire space of such mappings. Hence, LoRA adapters provide a parameter-efficient (typically, $m \cdot r + r \cdot n < m \cdot n$) way to adapt large models by learning only "important" directional updates confined to a subspace.

Moreover, many downstream adapters have been found to reuse the same "important" directions (Meng et al., 2024; Liu et al., 2024). We hypothesize that LoRA adapters may reuse *principal subspaces* that are fundamental for different domains of tasks. Once identified, task-specific weights can be found in these smaller subspaces rather than the whole weight space. To illustrate this idea clearly, we first define a space of tasks that are expressible using linear transformation matrices.

**Definition 1** (Linear Transformation Tasks). *Let $\mathcal{T} = \{t : x \in \mathbb{R}^n \to y \in \mathbb{R}^r\}$ denote a set of linear tasks where: $\forall\, t \in \mathcal{T}, \exists\, W_t \in \mathbb{R}^{r \times n}$ such that $y = W_t x + \epsilon_t$ , $\forall\, x, y$. Here, $\epsilon_t$ denotes the noise.*

A LoRA weight matrix at any layer does the same transformation. Without loss of generality, assume $r < n$ and let the transformation matrix $W_t \in \mathbb{R}^{r \times n}$ be interpreted as $r$ $n$-dimensional vectors: $\mathbf{w}_t^1, ..., \mathbf{w}_t^r \in \mathbb{R}^n$. Finding LoRA weights is equivalent to finding sets of these $r$ vectors in $\mathbb{R}^n$. Next, we define a subspace in $\mathbb{R}^n$.

**Definition 2** (Subspace). *Let $\mathcal{S}^{k,n} = \{\mathbf{a}_1, ..., \mathbf{a}_k\}$ ($k \leq n$) be a set of linearly independent vectors $\in \mathbb{R}^n$. Denote $\hat{\mathcal{S}}^{k,n} = span(\mathcal{S}^{k,n}) = \{\sum_{i=1}^k \alpha_i \mathbf{a}_i\ \forall i, \alpha_i \in \mathbb{R}\}$ as the subspace elicited by $\mathcal{S}^{k,n}$.*

Vectors in a subspace $\hat{\mathcal{S}}^{k,n}$ lie in $\mathbb{R}^n$ but are constrained to a smaller region. Similar to Tripuraneni et al. (2021), we use the following metric to measure distances between subspaces and vectors.

**Definition 3** (Distance between subspace and a vector). *Denote distance between a vector $\mathbf{v}$ and subspace $\hat{\mathcal{S}}^{k,n}$ as $\sin \theta(\mathbf{v}, \hat{\mathcal{S}}^{k,n})$, the sine of the principal angle $\theta$ between them. The principal angle is the smallest possible angle between a vector in the subspace and $\mathbf{v}$.*

Next, we introduce the idea of domain-specific subspaces.

**Definition 4** (Principal Subspace). *A subset of tasks $\mathcal{T}_d \subseteq \mathcal{T}$ constitutes a domain if, $\exists\, \mathcal{S}_d^{k,n}$, $\forall\, t \in \mathcal{T}_d$, such that $\sin \theta(\mathbf{w}_t^i, \hat{\mathcal{S}}_d^{k,n}) = 0\ \forall i \in 1, ..., r$. Denote $\hat{\mathcal{S}}_d^{k,n}$ as the principal subspace of $\mathcal{T}_d$.*

Here, $\sin \theta(\mathbf{w}_t^i, \hat{\mathcal{S}}_d^{k,n}) = 0$ implies that all the vectors constituting the weight matrix $W_t$ for all tasks $t$, lie inside the subspace spanned by $\mathcal{S}_d^{k,n}$. The existence of principal subspaces (PS) is trivially guaranteed for all domains $d$, e.g., when $k = n$. But, domains whose principal subspaces exist for $k \ll n$ would be practically useful. Even an *Approximate Principal Subspace* (APS), where the distance is small, i.e., $\sin \theta(\mathbf{w}_t^i, \hat{\mathcal{S}}^{k,n}) < \delta$ for some $\delta \approx 0$, would be useful, as we illustrate in Section 4. First, we present a theorem bounding the approximation error for recovering weights of new linear transformation tasks using a given APS characterized by $\delta$.

**Theorem 1.** *Given an APS* $(\hat{\mathcal{S}}_d^{k,n}; \delta)$, $\forall\, W_t \in \mathbb{R}^{r \times n}$ *of tasks* $t \in \mathcal{T}_d$, $\exists\, W_t' \in \hat{\mathcal{S}}_d^{k,n}$ *such that,*

$$\|W_t - W_t'\|_F < \delta\|W_t\|_F = \tan(\sin^{-1}\delta)\|W_t'\|_F \tag{1}$$

*Proof.* Let the weight matrix for task $t$, $W_t \in \mathbb{R}^{r \times n}$ be composed of vectors $\{\mathbf{w}_t^i\}_{i=1}^r$. By definition of APS, $\forall i, \sin\theta(\mathbf{w}_t^i, \hat{\mathcal{S}}_d^{k,n}) < \delta$. This implies that there exists a vector $\mathbf{w}_t^{i'} \in \hat{\mathcal{S}}_d^{k,n}$ such that $\sin\theta(\mathbf{w}_t^i, \mathbf{w}_t^{i'}) < \delta$, where $\mathbf{w}_t^{i'}$ is the projection of $\mathbf{w}_t^i$ on $\hat{\mathcal{S}}_d^{k,n}$ with an angle $\theta(\mathbf{w}_t^i, \mathbf{w}_t^{i'})$, or simply $\theta$ between them. Here, $\sin(\theta) = \frac{\|\mathbf{w}_t^i - \mathbf{w}_t^{i'}\|_2}{\|\mathbf{w}_t^i\|_2} < \delta$, and $\tan(\theta) = \frac{\|\mathbf{w}_t^i - \mathbf{w}_t^{i'}\|_2}{\|\mathbf{w}_t^{i'}\|_2} < \tan(\sin^{-1}\delta)$.

Then,

$$\|W_t - W_t'\|_F = \sqrt{\sum_{i=1}^r (\|\mathbf{w}_t^i - \mathbf{w}_t^{i'}\|_2)^2} < \sqrt{\sum_{i=1}^r (\delta\|\mathbf{w}_t^i\|_2)^2} = \delta\|W_t\|_F \quad \text{or,}$$

$$< \sqrt{\sum_{i=1}^r (\tan(\sin^{-1}\delta)\|\mathbf{w}_t^{i'}\|_2)^2} = \tan(\sin^{-1}\delta)\|W_t'\|_F$$

$\square$

Theorem 1 shows that for all task transformations that lie within the *principal subspace* of a domain, i.e. $\delta = 0$, we can recover them exactly using a linear combination of its principal components. For transformations outside this domain, i.e. $\delta \neq 0$, we can still find a transformation with bounded approximation error. In the worst case, when the transformation needs a component which is orthogonal to the principal subspace, i.e. $\delta = 1$, the approximation error can be unbounded (see Figure 2). Next, we present an algorithm to find principal subspaces using trained adapters and our experiments in Section 4 show that in most practical cases, the above approximation error is small.

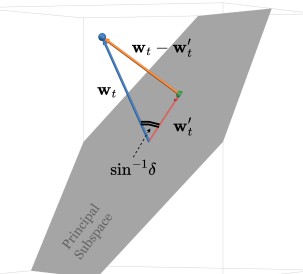

Figure 2: Projection on the principal subspace may incur an approximation error described by $\delta$.

### 3.2 ALGORITHM

Assume that we have $N$ LoRA adapters (sets of $A, B$ matrix pairs for each adapted layer), trained on various tasks in some domain $\mathcal{T}_d$, for some base pre-trained model $\mathcal{M}$. We present Algorithm 1 to calculate a list of principal components (eigenvectors that we call EigenLoRA PCs) which defines an approximate principal subspace (APS) for this domain. The algorithm stacks LoRA matrices (with variable ranks) at a particular layer to be analyzed for overlap. By treating each matrix as a list of vectors and decomposing this stack of vectors from across tasks, we find the most important components that can be linearly combined to approximate original weight matrices. We illustrate our algorithm using generic weight matrices ($W_t$). In practice, we apply the algorithm to all LoRA layer $A/B$ matrices.

---

**Algorithm 1 EigenLoRA PCs** extraction

**Input:** LoRA matrices $\{W_t \in \mathbb{R}^{r_t \times n}\}_{t=1}^N$, number of Principal (Eigen) Components ($K$)

**Output:** EigenLoRA PCs set $\mathcal{E}_d^{K,n}$, Mean $\hat{M}$ for translation.

$\hat{W} = \text{STACK}(\{W_t\}_{t=1}^N, \text{dim} = 0) \in \mathbb{R}^{R \times n}$     $\triangleright$ Stack all matrices. Here $R = \sum_t r_t$.
$M = \hat{W} - \hat{M}$     $\triangleright$ Zero-center them. Here $\hat{M} = \text{MEAN}(\hat{W}, \text{dim} = 0)$
$U, S, V^T = \text{SVD}(M)$     $\triangleright$ Perform Singular Value Decomposition.
$\mathcal{E}_d^{K,n} = V^T[: K]$     $\triangleright$ Choose top $K$ Eigen components.
return $\mathcal{E}_d^{K,n}, \hat{M}$

---

**Learning new adapters** Given a set of EigenLoRA PCs $\mathcal{E}_d^{K,n} = \{E_k \in \mathbb{R}^{1 \times n}\}_{k=1}^K$ (denoted simply by $\mathcal{E} \in \mathbb{R}^{K \times n}$ from here), an approximation $W_t'$ to any task matrix $W_t$ can be found by optimizing:

$$\min_\alpha \|W_t - W_t'\|_F,$$

where $\alpha \in \mathbb{R}^{K \times r}$ is a coefficient matrix that linearly combines the $K$ EigenLoRA PCs in $r$ different ways producing $W_t' = \text{STACK}(\{\text{SUM}(\alpha_j E_k, \dim = 0) + \hat{M}\}_{j=1}^r, \dim = 0) \in \mathbb{R}^{r \times n}$. In fact, we can analytically compute $\alpha^T = (W_t - \hat{M})\mathcal{E}^T$ for any weights $W_t$ to find the least distant projection $W_t'$ (which minimizes the above objective) on the subspace spanned by $\mathcal{E}$. However, we do not know the weights $W_t$ for new tasks in advance. In LoRA, both $A$ and $B$ which have $r \cdot n$ number of parameters need to be learned. But using EigenLoRA PCs, we can learn $\alpha$ instead. This replaces the original LoRA computation

$$h = W_0 x + BA(x) \quad \text{with} \quad h = W_0 x + \boxed{\alpha_B^T \mathcal{E}_B \alpha_A^T \mathcal{E}_A(x)}.$$

Here, $W_0$ are the pre-trained weights and $\mathcal{E}_B, \mathcal{E}_A$ are EigenLoRA PCs that remain fixed during training. The corresponding $\alpha_B^T$ and $\alpha_A^T$ are learned. This reduces the number of learnable parameters from $O(rn)$ to $O(rK)$, by a factor of $\frac{n}{K}$ (assuming rank $r$ to be fixed, which could also be changed). This provides a trade-off between subspace coverage (higher $K$) and parameter efficiency (increases learnable parameters).

**How to choose $K$?** The number of EigenLoRA PCs to be extracted is a hyperparameter chosen on the basis of *diversity* of tasks. The more *aligned* the weight matrices of a domain are, the fewer EigenLoRA PCs we need to achieve a low approximation error. However, this also restricts the space of weight matrices this set of EigenLoRA PCs could represent. More diverse weight matrices would need a higher number of EigenLoRA PCs to represent them, with the advantage of being able to represent a bigger space of tasks. A practical way to quantify the *diversity* of tasks is to look at the singular values of the EigenLoRA PCs. In Figure 3, we show a case where most of the information is contained in a handful of top EigenLoRA PCs. The percentage of cumulative singular values can be used as a threshold to decide $K$. More empirically, performance of reconstructed weight matrices on a validation set of tasks can be used to decide a suitable $K$.

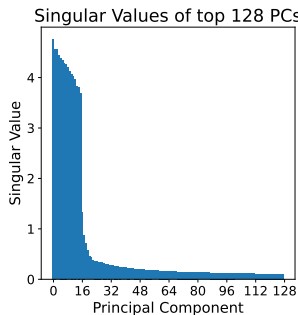

Figure 3: The top 16 components are most information dense ($A$ matrices from layer 1, LoRAHub, see Section 4.2.2).

**Memory-efficient Inference** In an application where multiple adapters are used (for example, image generation in multiple styles like photorealistic, sketch, etc.) frequently swapping between LoRA adapters adapters can be expensive. Either all adapters need to be loaded in GPU memory together (requiring larger memory) or they need to be loaded from CPU memory/disk (slow). With EigenLoRA, the EigenLoRA PCs are task-invariant while task-specific coefficients are lighter weight, allowing for more efficiency. For $N$ LoRAs, the memory footprint is $O(Nrn)$. For EigenLoRAs, it is $O(Kn + NrK)$. As $r, K \ll n$, EigenLoRA becomes $\frac{n}{K}$ times more memory efficient asymptotically. For example, serving $N = 8$ LoRAHub adapters (Section 4.2.2) would require $\approx 5\times$ less adapter memory than LoRA. This would be especially beneficial for mobile devices with small memory.

**Extreme Parameter Efficiency** Instead of stacking LoRA weight matrices, we can flatten them and treat them as vectors. In this case, the EigenLoRA (we call it EigenLoRA$^{\text{flat}}$) PCs are $r \cdot n$ dimensional (instead of $n$) and $\alpha$ is a $K$ dimensional vector that linearly combines these EigenLoRA PCs to produce an approximation of our weight matrix. This results in an additional reduction factor $r$ (the rank of each adapter needs to be fixed) in the number of parameters to learn but comes at an expense of some more model memory. This is analogous to a mixture-of-experts setting (where experts are EigenLoRA$^{\text{flat}}$ PCs). More details are deferred to Appendix A.2.2.

# 4    EXPERIMENTS AND ANALYSIS

In this section, we demonstrate the efficacy and versatility of EigenLoRA in a wide range of task complexities, modalities, and model architectures. We evaluated EigenLoRA on a simpler *aligned* domain setting with image classification tasks (Section 4.1); and a difficult *diverse* domain setting with NLP tasks (Section 4.2). We show that EigenLoRA requires substantially fewer parameters to achieve parity with or even exceed the performance of LoRA (Tables 1, 2, 3). Furthermore, it achieves loss convergence at a similar or faster rate (Figure 4), serving as a cost-effective alternative to random initialization (LoRA) and other existing initialization methods (Meng et al., 2024). Finally, we illustrate its memory-efficient inference capabilities with a text-to-image generation Stable Diffusion model (Rombach et al., 2021) (Section 4.3).

## 4.1    ALIGNED DOMAIN – IMAGE CLASSIFICATION

In this experiment, we test EigenLoRA with a pre-trained Vision Transformer (ViT) (Dosovitskiy et al., 2021) which is adapted for image classification on 3 data sets. The datasets are randomly divided into 5-6 sub-datasets with no overlap in categories, similar to continual learning (Kaushik et al., 2021) and federated learning (Shenaj et al., 2023) setups. Since the sub-datasets originate from a common dataset, their tasks are more *aligned* corresponding to the case where $\delta$ is small (see Section 3.1). For adaptation, we used LoRA (Hu et al., 2021) and VeRA (Kopiczko et al., 2023) to compare with our EigenLoRA. For completeness, we evaluated each method under multiple settings and report the mean performance across all sub-datasets.

**Setup**    We used the Huggingface PEFT library Mangrulkar et al. (2022) for the implementations for LoRA and VeRA, and followed their respective hyperparameter recommendations to train adapters for each sub-dataset from scratch. For EigenLoRA, we use all but one LoRA trained on individual sub-datasets to calculate EigenLoRA PCs (Algorithm 1) (leave-one-out). We then learn the coefficient matrix $\alpha$ for the left-out task using the method described in Section 3.2. Each method is finetuned for 10 epochs. Other experimental details are available in Appendix A.1.

**Parameter Efficiency**    Table 1 summarizes the results of our experiment. Note that all models require training of the last linear layer (with $\approx$15K parameters) since the pre-trained ViT has a different number of categories. For the Base Model, no other parameter is trained. For other models, some additional parameters are trained. EigenLoRA is capable of adapting to new sub-datasets using **only two principal components (or 96 additional trainable parameters)**. In fact, this small number of additional parameters for EigenLoRA help it match or outperform both LoRA and VeRA (both with considerably higher number of parameters). Lastly, we tested zero-shot EigenLoRA weights initialized randomly within the principal subspace and trained only the last layer (like the base model). The performance of this model exceeds that of the base model with no additional parameters, highlighting the effectiveness of extracting the principal subspaces. The list of trainable parameters and more details are available in appendix A.1.

Table 1: Aligned domain image classification with Vision Transformer. ZS refers to zero-shot. EigenLoRA matches or increases performance with drastically fewer number of parameters.

| | # Trainable Parameters | CIFAR100 | Food101 | Flowers102 | RESISC45 |
|---|---|---|---|---|---|
| Full Training | 86M | 97 | 96.64 | 98.82 | - |
| Base Model | 15K | 90.07 | 90.8 | 80.71 | 92.62 |
| LoRA 18 ($r = 4$) | +147K | 93.79 | **95.73** | 95.03 | **95.79** |
| LoRA ($r = 1$) | +36K | 92.45 | 91.07 | 90.14 | - |
| VeRA 24 | +18K | 90.87 | 91.75 | 91.25 | 92.81 |
| EigenLoRA ($K = 2$) | **+96** | **94.8** | 95.14 | **98.44** | 95.40 |
| EigenLoRA$^{ZS}$ | +0 | 91.4 | 92.48 | 95.7 | - |

## 4.2 DIVERSE DOMAIN – NATURAL LANGUAGE UNDERSTANDING

### 4.2.1 GLUE BENCHMARK

Next, we evaluate EigenLoRA on the General Language Understanding Evaluation (GLUE) benchmark (Wang et al., 2019) datasets using the RoBERTa$_{base}$ model (Liu et al., 2019). We use 6 different tasks: MRPC, SST-2, CoLA, QNLI, RTE and STS-B. Following the setup of VeRA, we omit time-intensive MNLI and QQP tasks, thus avoiding the use of MNLI initialization for MRPC, RTE, and STS-B tasks. In this setting, LoRAs are trained not on sub-datasets but on these different datasets representing the *diverse* domain setting, where $\delta$ may be larger than in the *aligned* domain setting. We follow the previous leave-one-out evaluation setup, where EigenLoRA PCs are calculated using LoRAs of all but one task, and $\alpha$ is learnt for the left-out task. Refer to Appendix A.2.1 for all hyperparameters and implementation details.

**Faster Convergence**  Our findings in Table 2 indicate that similar to the *aligned* domain experiments, EigenLoRA ($K = 32$) is able to match LoRA performance with $100\times$ **fewer trainable parameters**, while outperforming VeRA. EigenLoRA can effectively extract a useful principal subspace even from diverse domains and robustly adapt to new domains. In this setup, we also evaluate the weight initialization speed-up capability of EigenLoRA. This was recently studied by Meng et al. (2024) (PiSSA) who initialize their LoRA matrices with the principal directions of the pre-trained weight matrix ($W_0$). In contrast, we randomly initialize weights in our extracted principal subspace and compare its training convergence with other methods. The training loss graphs in Figure 4 demonstrate that **EigenLoRA achieves faster convergence than PiSSA and VeRA** and is slightly faster than LoRA, underscoring the importance of our extracted principal subspace. The reason for VeRA's poor performance as well as convergence maybe due to random initialization. It can be hard to optimize these random yet fixed weight components that may not align with task-critical principal components.

| Method | # Trainable Parameters | MRPC | SST-2 | CoLA | QNLI | RTE | STS-B | Avg. |
|---|---|---|---|---|---|---|---|---|
| Full Training | 125M | 88.97 | 91.28 | 59.81 | 92.29 | 79.78 | 90.89 | 83.84 |
| PISSA [37] | 1.2M | 86.52 | 94.15 | 61.32 | 92.15 | 71.84 | 90.25 | 82.70 |
| EigenLoRA$^{init}$ | 1.2M | 89.71 | 93.35 | 61.58 | 92.2 | 74.73 | 89.56 | 83.52 |
| LoRA ($r = 32$) | 1.2M | **86.76** | **94.72** | 59.56 | 92.53 | 77.61 | **90.81** | **83.67** |
| VeRA ($r = 256$) | 25K | 75.98 | 93.23 | 54.14 | 89.21 | 66.78 | 87.03 | 77.72 |
| EigenLoRA | **12K** | 87 | 94.15 | **59.81** | **92.73** | **77.62** | 90.58 | 83.65 |

Table 2: GLUE benchmark results. We report Matthew's correlation for CoLA, Pearson correlation for STS-B, and accuracy for the remaining tasks. In all cases, higher values indicate better performance.

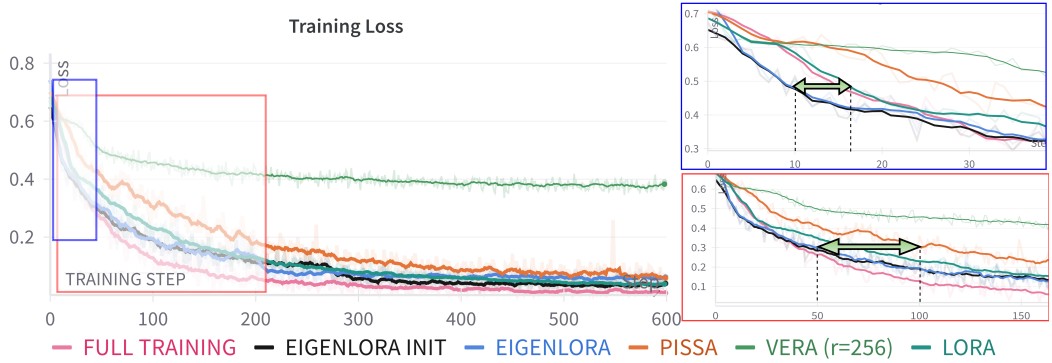

Figure 4: **Fast Covergence and Better Initialization** EigenLoRA demonstrates fast convergence compared to LoRA and VeRA. EigenLoRA achieves a speedup of up to $1.5\times$ against LoRA and up to $2\times$ compared to PISSA. This experiment was carried out on the CoLA task of the GLUE benchmark.

### 4.2.2 LoRAHub

Finally, we also tested our method in settings where the adapters may be trained on significantly diverse domains. LoRAHub (Huang et al., 2023) is a collection of 196 adapters of the FLAN T5 Large model (Chung et al., 2024), trained on a variety of task domains like Reading Comprehension (Adversarial QA (Bartolo et al., 2020), DuoRC (Saha et al., 2018), etc.), Text Classification (BoolQ (Clark et al., 2019), etc.), Math (Hendrycks et al., 2021), Text Generation (Maas et al., 2011), etc. LoRAHub represents the realistic setting where we directly use publicly available trained adapters, which may present significant diversity in terms of quality and task domain.

**Setup**  Not all publicly available adapters are useful. After filtering out bad adapters (see Appendix A.2.2), we were left with 68 adapters, where the performance of the LoRA model exceeded base model substantially. As running leave-one-out experiments are expensive, we split the 68 adapters randomly into two sets $(53, 15)$. EigenLoRA PCs were calculated using the larger "training" set and evaluations were done on the smaller "test" set. We evaluated EigenLoRA under different settings: EigenLoRA$^{\text{flat}}$ (Section 3.2) for extreme parameter efficiency, zero-shot (ZS) (randomly selecting weights from the principal subspace) and Analytical reconstruction (AL) (calculated using the already available adapter weights, no training). The performance on two individual datasets along with the average across the 15 test domains is reported in Table 3. Some other results are defered to Appendix A.2.2.

**EigenLoRA outperforms LoRA with $32\times$ fewer parameters**. In fact, the smallest possible LoRA with $r = 1$ still uses $2\times$ more parameters than EigenLoRA while gaining no performance over the base model. Zero-shot results highlight the significance of identifying the principal subspace. Even randomly selected weights within that subspace achieve better performance than base model. Although EigenLoRA$^{\text{flat}}$ is memory-extensive when training, it uses even fewer number of parameters and achieves similar performance. Finally, the analytically calculated EigenLoRA weights represent the projection of original LoRA weights on the identified principal subspace. Our trained EigenLoRAs reach close to or even surpass the performance of these weights showing that $\alpha$ can be easily optimized.

Table 3: Evaluation of our methods on LoRAHub (Diverse domain).

| | # Trainable Parameters | Amazon Review Polarity | Wiki Generate Subject | Average (15 tasks) |
|---|---|---|---|---|
| Base Model | 0 | 34.02 | 9.03 | 50.83 |
| LoRA ($r = 16$) | 4.7M | **96.18** | 39.97 | 63.10 |
| LoRA ($r = 1$) | 295K ($\downarrow 16\times$) | 34.02 | 9.03 | 50.83 |
| EigenLoRA ($K = 32$) | 147K ($\downarrow 32\times$) | **96.18** | **40.97** | **63.50** |
| EigenLoRA$^{\text{flat}}$ ($K = 8$) | 2K ($\downarrow 2400\times$) | 96.18 | 11.37 | 60.52 |
| EigenLoRA$^{\text{ZS}}$ | 0 | 39.59 | 9.03 | 51.29 |
| EigenLoRA$^{\text{ZS-flat}}$ | 0 | 69.16 | 9.03 | 53.50 |
| EigenLoRA$^{\text{AL}}$ ($K = 32$) | 0 | 96.66 | 38.63 | 64.04 |

### 4.3 Memory-efficient Inference – Text-to-Image Models

As adapters become more common, we see a new challenge in efficiently hosting multiple adapters at the same time for different tasks. An example application domain is image generation, where multiple adapters correspond to different generation styles. If we want to quickly change between styles, we would need to swap an active adapter with another, potentially from CPU memory or disk. This can significantly slow down inference and can be performance critical in edge devices. We know that EigenLoRA can reduce the number of in-memory parameters by extracting and reusing a task-invariant subspace. Instead of using EigenLoRA to train new adapters, we can also use it to perform memory-efficient inference.

**Analytical Reconstruction**  To show EigenLoRA's efficacy, we extracted $K = 14$ EigenLoRA PCs from $N = 20$ Stable Diffusion-XL (Podell et al., 2023) LoRA adapters (rank $r = 32$) taken

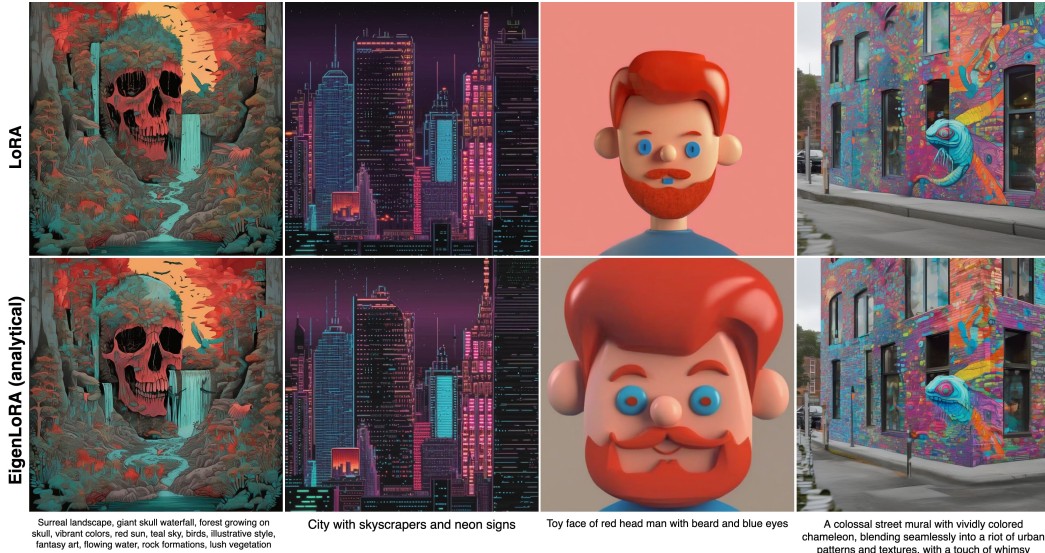

Figure 5: Analytical reconstruction of LoRAs using EigenLoRA can result in substantial reduction in memory usage without much degradation in visual quality. See Appendix A.3 for more examples.

from HuggingFace diffusers library von Platen et al. (2022). We use $r \in \{16, 32\}$ in $\alpha \in \mathbb{R}^{r \times K}$ to analytically calculate the projected weights of original LoRAs on the extracted principal subspace. The number of denoising steps during image generation was set to 30 and the seed was set to 0. Images from these EigenLoRAs and their corresponding original LoRAs can be seen in Figure 5. This reconstruction reduces the number of parameters to store all adapters from 4.6GB to 261MB. **This results in approximately $18\times$ reduction in number of low-rank parameters needed to be stored in memory**. This is significant, especially if the LoRA size and number is large . With EigenLoRA, a large number of adapters can be stored at once in GPU memory and easily swapped.

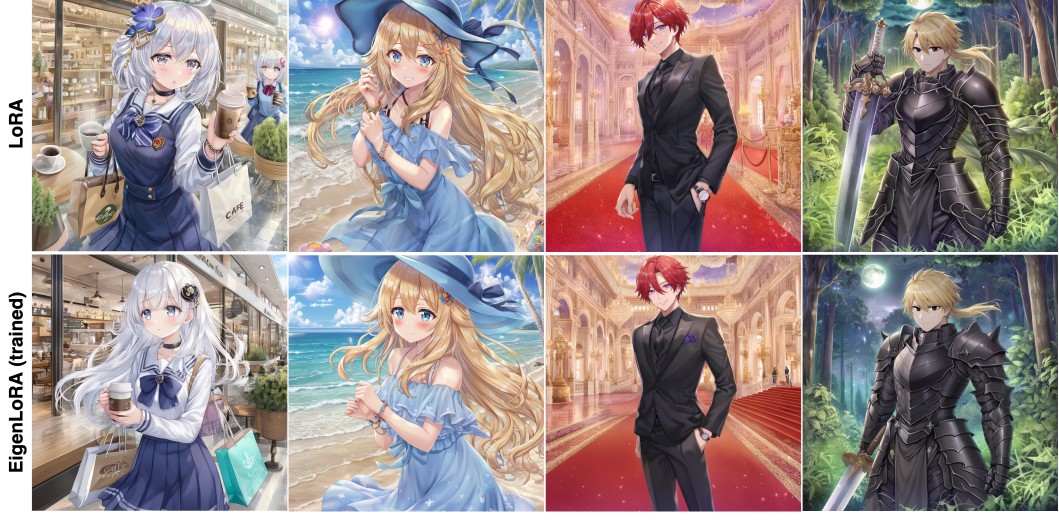

Figure 6: Comparison of images generated by LoRA and EigenLoRA trained on Torino Aqua anime style images. For EigenLoRA, we utilized 12 components with only trainable coefficients to finetune the base model.

**Trained EigenLoRAs**    Lastly, we also show the results of trainable EigenLoRAs in this domain. In this setup, we use a version of Stable-Diffusion-XL 1 model as our base model and use publicly available LoRA adapters from the HuggingFace diffusers (von Platen et al., 2022) repository which

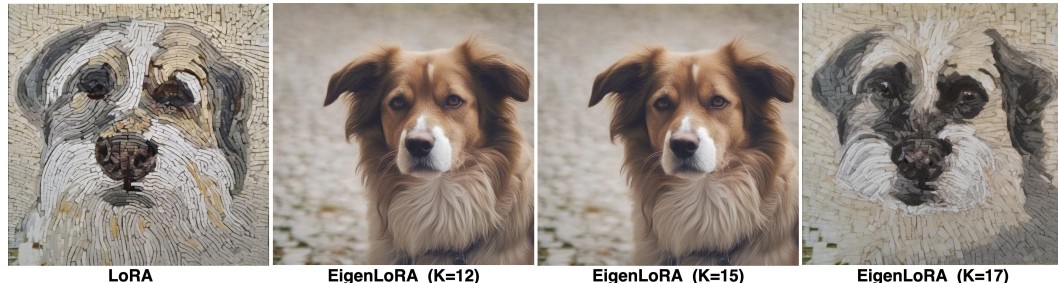

| LoRA | EigenLoRA (K=12) | EigenLoRA (K=15) | EigenLoRA (K=17) |

Figure 7: Failure Case: EigenLoRA may struggle if a task-critical principal component is not present in the extracted principal subspace. In this case, the model loses the important "mosaic" property in the generated image for the prompt: "mosiac picture of a dog".

have been trained on different anime styles to extract the EigenLoRA PCs.. We train coefficients for $K = 12$ EigenLoRA PCs to adapt the model to a new domain using publicly available digital anime art data by a digital artist, Torino Aqua, whose work is defined by a unique blend of colorful palettes, intricate linework, and expressive character designs. The objective is to generate images in the similar artistic style. We show the results in Figure 6. We compare the results of a LoRA and our EigenLoRA (K=12) trained on similar data on the same base model 1 and observe that EigenLoRAs are able to get visual quality similar to LoRA at a fraction of the compute cost.

## 5  CONCLUSION, LIMITATIONS AND OPPORTUNITIES

We introduce EigenLoRA, a PEFT method that recycles trained adapters by finding a task-invariant principal subspace. This allows for more efficient training of new adapters and inference with multiple adapters. Through experiments, we showed that EigenLoRA works and provides practical benefits in a wide range of scenarios. Our method has the potential to mitigate the perpetually widening compute resource gap (Ahmed & Wahed, 2020; Besiroglu et al., 2024) and reduce the environmental cost of training and using AI models (Wu et al., 2021; Ligozat et al., 2021). It also holds promise for training personalized models (Tan et al., 2024) on low-resource devices, in privacy-critical use-cases.

However, there are some potential limitations of our method. Figure 7 presents a failure case, where it fails to achieve a key property of the desired image. As mentioned in Section 3.1, the approximation error in a subspace projection depends on components orthogonal to that subspace, even if all tasks may share a principal subspace. If these orthogonal components are critical for a task, performance will suffer. This is because EigenLoRA does not search for weights outside of the principal subspace. However, a simple extension of EigenLoRA which frees a small number of rank-1 weights to be trainable outside of the principal subspace, can avoid this problem. This would change the EigenLoRA computation from $h = W_0 x + \alpha_B^T \mathcal{E}_B \alpha_A^T \mathcal{E}_A(x)$ to say, $h = W_0 x + (\alpha_B^{:-1})^T \mathcal{E}_B^{:-1} (\alpha_A^{:-1})^T \mathcal{E}_A^{:-1}(x) + B_1 A_1(x)$, where $\mathcal{E}_B^{:-1}, \mathcal{E}_A^{:-1}$ represent top $K-1$ fixed EigenLoRA PCs, $\alpha_B^{:-1}, \alpha_A^{:-1}$ their respective learnable coefficients and $B_1, A_1$ represent rank-1 free learnable weights. Moreover, our experiments do not include empirical optimizations at each layer or individual weight matrix level. Although we experimented with different values of $K$, it was fixed for all layers and both $A, B$ matrices in each experiment. This can be further optimized empirically as discussed in Section 3.2. Lastly, EigenLoRA$^{\text{flat}}$ has potential to be used as a mixture-of-experts model. We defer these extensions and optimization for future work.

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

## A APPENDIX

### A.1 EXPERIMENTS

For VeRA, LoRA and PiSSA, we experimented with a range of learning rates, from higher to lower, along with three different scheduling approaches: ReduceLRonPlateau, Linear, and Cosine. The hyperparameters that yielded the best average performance were selected for further experimentation.

The observed discrepancies with EigenLoRA hyperparameters are attributable to these methodological choices. Comprehensive hyperparameter tuning for EigenLoRA was not pursued extensively, as the initially selected hyperparameters, notably a high learning rate paired with ReduceLRonPlateau or Linear, demonstrated satisfactory performance, thereby conserving computational resources.

### A.1.1   IMAGE CLASSIFICATION

**Trainable parameters for EigenLoRA**   The base model is vit-base-patch16-224. The following are the trainable parameters in ViT (Dosovitskiy et al., 2021) that are trained for EigenLoRA. We ignore the last linear layer for simplicity since it is trained for all models and baselines and is constant. The loading parameter has the shape of [number of EigenLoRA PC, 1] (we only have 2 in each EigenLoRA PC for this experiment). Therefore, the total number of trainable parameters (for the number of components= 2) is 12 (layers) $\times$ 4 (set of parameters per layers) $\times$ 2 (number of trainable parameter per coefficient)  = 96 trainable parameters.

```
model.encoder.layer.0.attention.attention.query.eigenlora_A.loadings
model.encoder.layer.0.attention.attention.query.eigenlora_B.loadings
model.encoder.layer.0.attention.attention.value.eigenlora_A.loadings
model.encoder.layer.0.attention.attention.value.eigenlora_B.loadings
model.encoder.layer.1.attention.attention.query.eigenlora_A.loadings
model.encoder.layer.1.attention.attention.query.eigenlora_B.loadings
model.encoder.layer.1.attention.attention.value.eigenlora_A.loadings
model.encoder.layer.1.attention.attention.value.eigenlora_B.loadings
model.encoder.layer.2.attention.attention.query.eigenlora_A.loadings
model.encoder.layer.2.attention.attention.query.eigenlora_B.loadings
model.encoder.layer.2.attention.attention.value.eigenlora_A.loadings
model.encoder.layer.2.attention.attention.value.eigenlora_B.loadings
model.encoder.layer.3.attention.attention.query.eigenlora_A.loadings
model.encoder.layer.3.attention.attention.query.eigenlora_B.loadings
model.encoder.layer.3.attention.attention.value.eigenlora_A.loadings
model.encoder.layer.3.attention.attention.value.eigenlora_B.loadings
model.encoder.layer.4.attention.attention.query.eigenlora_A.loadings
model.encoder.layer.4.attention.attention.query.eigenlora_B.loadings
model.encoder.layer.4.attention.attention.value.eigenlora_A.loadings
model.encoder.layer.4.attention.attention.value.eigenlora_B.loadings
model.encoder.layer.5.attention.attention.query.eigenlora_A.loadings
model.encoder.layer.5.attention.attention.query.eigenlora_B.loadings
model.encoder.layer.5.attention.attention.value.eigenlora_A.loadings
model.encoder.layer.5.attention.attention.value.eigenlora_B.loadings
model.encoder.layer.6.attention.attention.query.eigenlora_A.loadings
model.encoder.layer.6.attention.attention.query.eigenlora_B.loadings
model.encoder.layer.6.attention.attention.value.eigenlora_A.loadings
model.encoder.layer.6.attention.attention.value.eigenlora_B.loadings
model.encoder.layer.7.attention.attention.query.eigenlora_A.loadings
model.encoder.layer.7.attention.attention.query.eigenlora_B.loadings
model.encoder.layer.7.attention.attention.value.eigenlora_A.loadings
model.encoder.layer.7.attention.attention.value.eigenlora_B.loadings
model.encoder.layer.8.attention.attention.query.eigenlora_A.loadings
model.encoder.layer.8.attention.attention.query.eigenlora_B.loadings
model.encoder.layer.8.attention.attention.value.eigenlora_A.loadings
model.encoder.layer.8.attention.attention.value.eigenlora_B.loadings
model.encoder.layer.9.attention.attention.query.eigenlora_A.loadings
model.encoder.layer.9.attention.attention.query.eigenlora_B.loadings
model.encoder.layer.9.attention.attention.value.eigenlora_A.loadings
model.encoder.layer.9.attention.attention.value.eigenlora_B.loadings
model.encoder.layer.10.attention.attention.query.eigenlora_A.loadings
model.encoder.layer.10.attention.attention.query.eigenlora_B.loadings
model.encoder.layer.10.attention.attention.value.eigenlora_A.loadings
model.encoder.layer.10.attention.attention.value.eigenlora_B.loadings
model.encoder.layer.11.attention.attention.query.eigenlora_A.loadings
model.encoder.layer.11.attention.attention.query.eigenlora_B.loadings
model.encoder.layer.11.attention.attention.value.eigenlora_A.loadings
model.encoder.layer.11.attention.attention.value.eigenlora_B.loadings
```

**Hyperparameters** LoRA (Hu et al., 2021) and VeRA (Kopiczko et al., 2023) implementations are taken from the HuggingFace PEFT (Mangrulkar et al., 2022) library with hyperparameters of the default method. For Food101 (Bossard et al., 2014) experiment, we randomly remove 1 class for ease of compute. Experimental hyperparameters are reported in Table 4 and Table 5.

|  | CIFAR100 | Flowers102 | Food101 | RESISC45 |
|---|---|---|---|---|
| Learning Rate | 1e−4 | 1e−4 | 1e−4 | 1e−3 |
| Weight Decay | 0.1 | 0.1 | 0.1 | 0.1 |
| Warmup ratio | 0.06 | 0.06 | 0.06 | 0.06 |
| Epochs | 10 | 10 | 10 | 10 |
| Number of Subsets | 5 | 6 | 5 | 5 |
| Categories/Subset | 20 | 17 | 20 | 9 |
| Seed | 42 | 42 | 42 | 42 |
| Batch Size | 128 | 64 | 128 | 128 |

Table 4: Hyperparameters for LoRA (Hu et al., 2021) and VeRA (Kopiczko et al., 2023) for the Image Classification Experiment

|  | CIFAR100 | Flowers102 | Food101 | RESISC45 |
|---|---|---|---|---|
| Learning Rate | 1e−2 | 1e−2 | 1e−2 | 1e−3 |
| Weight Decay | 0.1 | 0.1 | 0.1 | 0.1 |
| Warmup ratio | 0.06 | 0.06 | 0.06 | 0.06 |
| Epochs | 10 | 10 | 10 | 10 |
| Number of Subsets | 5 | 6 | 5 | 5 |
| Categories/Subset | 20 | 17 | 20 | 9 |
| Seed | 42 | 42 | 42 | 42 |
| Batch Size | 128 | 64 | 128 | 128 |

Table 5: Hyperparameters for EigenLoRA for the Image Classification Experiment

**Experimental Results** The experiments were conducted 5 times utilizing randomly generated dataset splits. The mean accuracy values are reported in Table 1. Empirical analysis indicates that without control and annealing of learning rates, the loss for both LoRA and VeRA may diverge or plateau, particularly with high learning rates. Even with the lower learning rate, Full training or LoRA can overfit to the training data without proper regularization. In contrast, no such instability was observed during EigenLoRA training, where a relatively higher learning rate proved advantageous for rapid convergence.

| Model | Trainable Params | subset1 | subset2 | subset3 | subset4 | subset5 | Avg. |
|---|---|---|---|---|---|---|---|
| **FT** | 86389248 | 98.8 | 97.95 | 95.55 | 96.05 | 96.3 | 96.93 |
| **LoRA** ($r = 1$) | 36864 | 97.6 | 93.95 | 93.75 | 91.75 | 85.2 | 92.45 |
| **LoRA** ($r = 4$) | 147456 | 98.15 | 95.2 | 93.5 | 92.85 | 89.25 | 93.79 |
| **VeRA** ($r = 2$) | 18480 | 93.65 | 89.7 | 89.5 | 89.95 | 91.55 | 90.87 |
| **EigenLoRA** ($K = 2$) | 96 | 97.25 | 95.05 | 94.55 | 93 | 94.15 | 94.8 |

Table 6: Image Classification Accuracy results on CIFAR100 (Krizhevsky et al., 2009)

| Model | Trainable Params | subset1 | subset2 | subset3 | subset4 | subset5 | Avg. |
|---|---|---|---|---|---|---|---|
| **FT** | 86389248 | 98.64 | 97 | 97.36 | 94.28 | 95.92 | 96.64 |
| **LoRA** ($r = 1$) | 36864 | 93.36 | 88.44 | 94.28 | 89.4 | 89.9 | 91.076 |
| **LoRA** ($r = 4$) | 147456 | 98.2 | 96.96 | 96.08 | 92.88 | 94.52 | 95.728 |
| **VeRA** ($r = 2$) | 18480 | 91.22 | 88.42 | 94.42 | 91.88 | 92.82 | 91.752 |
| **EigenLoRA** ($K = 2$) | 96 | 97.24 | 95.96 | 96 | 91.88 | 94.6 | 95.136 |

Table 7: Image Classification Accuracy results on Food101 (Bossard et al., 2014)

| Model | subset1 | subset2 | subset3 | subset4 | subset5 | subset6 | Avg. |
|---|---|---|---|---|---|---|---|
| **FT** | 99.7 | 99.3 | 98.01 | 98.22 | 99.7 | 98.01 | 98.82 |
| **LoRA** ($r = 1$) | 85.9 | 88.47 | 92.69 | 91.02 | 91.7 | 91.01 | 90.13 |
| **LoRA** ($r = 4$) | 96.23 | 92.76 | 97.22 | 95.01 | 98.24 | 90.73 | 95.03 |
| **VeRA** ($r = 2$) | 99.2 | 95.4 | 97.7 | 94.7 | 90.9 | 95 | 95.48 |
| **EigenLoRA** ($K = 2$) | 99.686 | 97.905 | 97.689 | 98.291 | 99.344 | 97.718 | 98.43 |

Table 8: Image Classification Accuracy results on Flowers102 (Nilsback & Zisserman, 2008)

## A.2 NATURAL LANGUAGE PROCESSING

### A.2.1 NLU - GLUE BENCHMARK

**Hyperparameters** LoRA (Hu et al., 2021), VeRA (Kopiczko et al., 2023) and PISSA (Meng et al., 2024) implementations are taken from the HuggingFace PEFT (Mangrulkar et al., 2022) library. Refer to Table 9 and Table 10 for hyperparameter details. For LoRA (Hu et al., 2021), we use the ranks $\in \{8, 16\}$. For VeRA (Kopiczko et al., 2023), we use rank$= 256$, and for EigenLoRA, we use $K \in \{16, 32\}$ and $r = 8$. Here, $r$ refers to the dimensionality of the trainable coefficients and not the rank. For both PISSA (Meng et al., 2024) and LoRA, all the parameters of the low rank matrix are trainable. For the EigenLoRA initialization experiment, we train both the components and coefficients for a fair comparison with PISSA. In practice, however, we do not need to do so - we can tune only the sparse coefficients and after the loss converges, finetune the components for a few training steps.

| | CoLA | MRPC | QNLI | RTE | SST-2 | STSB |
|---|---|---|---|---|---|---|
| Learning Rate | 4e−4 | 4e−4 | 4e−4 | 5e−4 | 5e−4 | 4e−4 |
| Weight Decay | 0.1 | 0.1 | 0.1 | 0.1 | 0.1 | 0.1 |
| Warmup ratio | 0.06 | 0.06 | 0.06 | 0.06 | 0.06 | 0.06 |
| Epochs | 80 | 80 | 25 | 80 | 60 | 40 |
| Scheduler | Linear | Linear | Linear | Linear | Linear | Linear |
| Seed | 0 | 0 | 0 | 0 | 0 | 0 |
| Batch Size | 64 | 64 | 64 | 64 | 64 | 64 |

Table 9: Hyperparameters for LoRA (Hu et al., 2021), VeRA (Kopiczko et al., 2023) and PiSSA (Meng et al., 2024) for the GLUE benchmark. (Wang et al., 2019)

|  | CoLA | MRPC | QNLI | RTE | SST-2 | STSB |
|---|---|---|---|---|---|---|
| Learning Rate | $4e{-}3$ | $4e{-}3$ | $4e{-}3$ | $5e{-}3$ | $5e{-}3$ | $4e{-}3$ |
| Weight Decay | 0.1 | 0.1 | 0.1 | 0.1 | 0.1 | 0.1 |
| Warmup ratio | 0.06 | 0.06 | 0.06 | 0.06 | 0.06 | 0.06 |
| Epochs | 80 | 80 | 25 | 80 | 60 | 40 |
| Scheduler | RLrP | RLrP | RLrP | RLrP | RLrP | RLrP |
| Seed | 0 | 0 | 0 | 0 | 0 | 0 |
| Batch Size | 64 | 64 | 64 | 64 | 64 | 64 |

Table 10: Hyperparameters for EigenLoRA for the GLUE benchmark. (Wang et al., 2019) (RLrP - ReduceLRonPlateau)

### A.2.2 LORAHUB

For filtering LoRAHub adapters, we used a criterion of at least $2\%$ improvement in performance on adapter training data compared to base model. It is surprising that 128 of the 196 adapters did not qualify under this criteria. It is important to filter out such adapters because if some weights do not add anything meaningful to the base model, they might be noisy and in turn affect the extraction of good EigenLoRA PCs.

We conducted more experiments with variations of $K$ in both EigenLoRA ($K = 16, 32, 64, 128, 256$) and EigenLoRA$^{\text{flat}}$ ($K = 4, 8, 12, 16$). We found that EigenLoRA$^{\text{flat}}$ increased in performance with increasing $K$ but it is difficult to train these models due to excessive memory requirements. We also found that EigenLoRA performance peaked at $K = 32$ and remained similar for higher $K$, indicating the potential existence of noisy components that are not useful for adaptation. We present some of these extra results here in Table 11.

Table 11: Evaluation of our methods on LoRAHub (Diverse task domain).

|  | # Trainable Parameters | Amazon Review Polarity | Wiki Generate Subject | Average (15 tasks) |
|---|---|---|---|---|
| Base Model | 0 | 34.02 | 9.03 | 50.83 |
| LoRA ($r = 16$) | 4.7M | 96.18 | 39.97 | 63.10 |
| LoRA$^{\text{init}}$ ($r = 16$) | 4.7M | 96.34 | 39.80 | 63.24 |
| LoRA$^{\text{init-flat}}$ ($r = 16$) | 4.7M | 95.87 | 40.31 | 63.19 |
| LoRA ($r = 1$) | 295K ($\downarrow 16\times$) | 34.02 | 9.03 | 50.83 |
| EigenLoRA ($K = 32$) | 147K ($\downarrow 32\times$) | 96.18 | 40.97 | 63.50 |
| EigenLoRA$^{\text{ZS}}$ | 0 | 39.59 | 9.03 | 51.29 |
| EigenLoRA$^{\text{flat}}$ ($K = 8$) | 2K ($\downarrow 2400\times$) | 96.18 | 11.37 | 60.52 |
| EigenLoRA$^{\text{ZS-flat}}$ | 0 | 90.94 | 9.03 | 58.45 |
| EigenLoRA$^{\text{AL}}$ ($K = 32$) | 0 | 96.66 | 38.63 | 64.04 |
| EigenLoRA$^{\text{AL-flat}}$ ($K = 8$) | 0 | 96.34 | 23.91 | 62.80 |

### A.3 TEXT-TO-IMAGE GENERATION (STABLE DIFFUSION MODELS)

Figure 8 and Figure 9 show more examples of a text-to-image stable diffusion model finetuned using EigenLoRA. Note that not only there is no publicly available code for VeRA that allows its usage in complex text-to-image generation tasks, but our VeRA implementation also did not work well in this task.

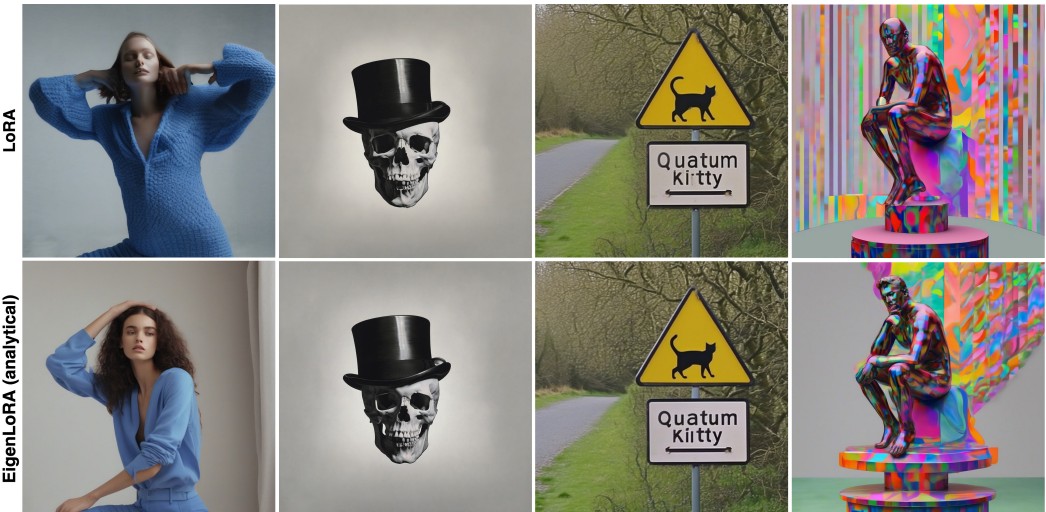

Figure 8: (Part 1) A single EigenLoRA (identical components, varying loadings) was employed to produce these images utilizing the Stable Diffusion-XL Podell et al. (2023) model. A comparison between our results and those obtained from multiple LoRAs does not show a noticeable degradation in visual quality.

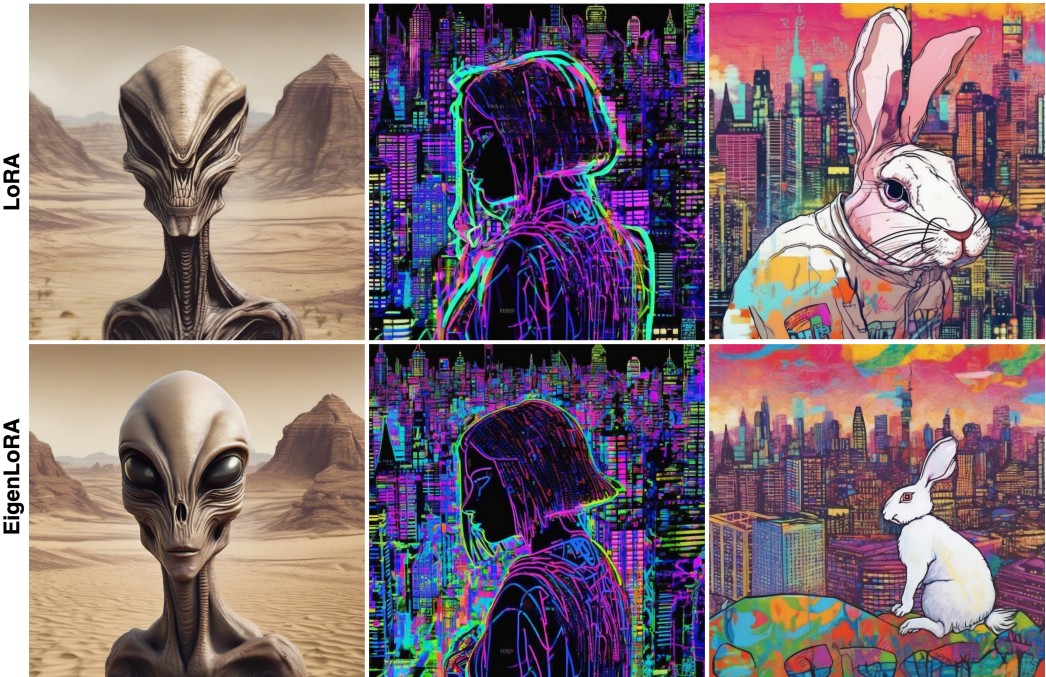

Figure 9: (Part 2) A single EigenLoRA (identical components, varying loadings) was employed to produce these images utilizing the Stable Diffusion-XL Podell et al. (2023) model. A comparison between our results and those obtained from multiple LoRAs demonstrates no noticeable degradation in visual quality.

## A.4 ADDITIONAL EXPERIMENTS

We present additional results from CommonSense Reasoning Table 12 and Arithmetic Reasoning Table 13 tasks using LLAMA-7B model (Hu et al., 2023). We follow the one-out strategy, in which we initialize EigenLoRA PCs using LoRAs learned on other available task LoRAs and then the

coefficients are finetuned using CommonSense170k and Math10k datasets (Hu et al., 2023). For LoRA, rank= 32 and for EigenLoRA, $k = 32$. The recommended hyperparameters were followed from (Hu et al., 2023).

| Method | Param | BoolQ | PIQA | SIQA | HellaSwag | WinoGrande | ARC-e | ARC-c | OBQA | Average |
|--------|-------|-------|------|------|-----------|------------|-------|-------|------|---------|
| LoRA | 56.1M | 68.53 | 71.22 | 67.91 | 80.91 | 73.64 | 72.13 | 56.74 | 62.2 | 69.16 |
| ELoRA | 0.3M | 68.55 | 72.05 | 71.1 | 78.1 | 74.24 | 71.13 | 56.7 | 64.54 | 69.55 |

Table 12: Accuracy comparison with different adapters on eight commonsense reasoning datasets.

| Method | Param | MultiArith | GSM8K | AddSub | AQuA | SingleEq | SVAMP | Avg. |
|--------|-------|------------|-------|--------|------|----------|-------|------|
| LoRA | 56.1M | 95 | 37.5 | 83.3 | 18.9 | 84.4 | 52.11 | 61.86 |
| ELoRA | 0.3M | 95.66 | 39.91 | 83.04 | 17.64 | 84.65 | 50.25 | 61.85 |

Table 13: Arithmetic Reasoning Results

In addition, we also performed a 3D object pose estimation (Angtian et al., 2021; Kaushik et al., 2024) finetuning experiment using a modified Resnet-101. The task of 3D object pose estimation involves the prediction of three rotation parameters (azimuth, elevation, in-plane rotation) of an object relative to the camera. The pose estimation error between the predicted rotation matrix and the ground truth rotation matrix is given as $\Delta(R_{pred}, R_{gt}) = \frac{|| \log_b(R_{pred}^{\mathsf{T}} R_{gt})||_F}{\sqrt{2}}$ We show the results for the $\frac{\pi}{6}$ accuracy threshold for this experiment.

| Method | Param | Airplane | Motorbike | Boat | Bottle | Bus | Car | Average |
|--------|-------|----------|-----------|------|--------|-----|-----|---------|
| LoRA (r=16) | 215K | 79.9 | 80.1 | 71.5 | 89.8 | 90.1 | 96.6 | 84.67 |
| VeRA (r=256) | 40K | 68.4 | 72.4 | 64.3 | 88.4 | 87.2 | 94.4 | 79.18 |
| EigenLoRA (K=2) | 16K | 81.4 | 80.0 | 71.4 | 90 | 92.3 | 97.5 | 85.43 |

Table 14: 3D object pose estimation accuracy ($\frac{\pi}{6}$ threshold)

# B METHOD ANALYSIS AND ABLATION

Through a rigorous comparative analysis of EigenLoRAs and their target LoRAs, we identified that the most pronounced reconstruction discrepancies manifest in the initial and terminal layers of the neural network, as depicted in Figure 10. Allowing the EigenLoRA PCs in these layers to undergo fine-tuning alongwith the coefficients can alleviate failure scenarios, thereby alleviating the need for comprehensive model fine-tuning.

## B.1 HOW TO CHOOSE $K$ PRINCIPAL COMPONENTS AND $r$ FOR EIGENLORA

We perform an ablation study on the selection of EigenLoRA principal components (K). Our analysis concentrates on one experiment as shown in Figure 11, specifically pertaining to the MRPC task within the GLUE (Wang et al., 2019) benchmark. The analysis in Figure 11a shows the training loss in relation to increasing number of EigenLoRA principal components $K$, as well as the explained variance of the LoRAs used to initialize the EigenLoRA in Figure 11b. We find, empirically, that choosing EigenLoRA PCs for explained variance of $50 - 80\%$ of the LoRAs used to initialize the EigenLoRA is sufficient for a robust initialization. This is shown in fig. 11b where we choose $K = 8$ which roughly corresponds to the explained variance of $55 - 60\%$. We further ablate this choice in fig. 11a, where although substantial improvements are evident up to $K = 8$, an increase in the number of $K$ thereafter yields only marginal gains, demonstrating diminishing returns as the number of components increases. The parameter $r$ in EigenLoRA does not equate the *rank* parameter in LoRA and its variants. It reflects the dimensionality of the EigenLoRA coefficients. Although $r = 1$ works well, we observe slight performance improvements as we increase this value as shown in fig. 12. Increasing this value corresponds to a small amount of parameter increase. We observe no

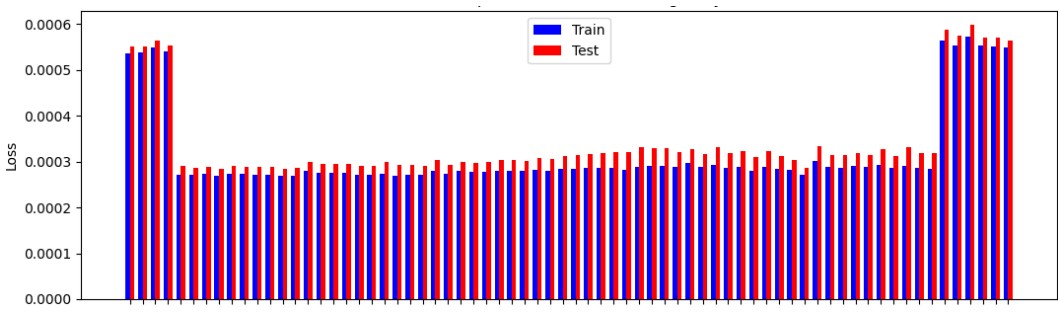

Figure 10: Average reconstruction error between EigenLoRA and a set of LoRAs for all UNet layers in a stable diffusion model.

finetuning instability by changing this value and recommend that it can be set to anywhere between 1 and the rank of the LoRAs used to initialize EigenLoRA.

## B.2 FAILURE CASES

Figure 7 illustrates a potential failure case of EigenLoRA, where the incorrect number of principal components (PCs) was selected. In this instance, the "mosaic style" information was excluded from the principal subspace identified by EigenLoRA due to an insufficient number of PCs. However, this issue can be resolved by selecting a larger number of PCs, as the extended principal subspace contains the necessary information for the task.

Another hypothetical failure scenario arises if the domain gap between the low-rank adapters used to initialize EigenLoRA and the downstream task is significantly large. Although we do not observe such a case in our experiments, it is plausible that under such conditions, EigenLoRA might underperform. This issue could potentially be mitigated by allowing only a subset of PCs to remain trainable, enabling the model to adapt more effectively to the target domain.

A further observed limitation of EigenLoRA occurs in complex tasks like Text-to-Image generation, which may extend to other tasks as well. If the majority of LoRAs used to initialize EigenLoRA encode biases (e.g., related to gender, race, or context), these biases tend to propagate into EigenLoRA outputs. While such biases are a common issue in deep learning models trained using stochastic gradient descent or similar methods, addressing them remains a critical area of future work. We consider this an important avenue for improvement and discuss the broader implications in appendix C.

## B.3 IMPACT OF LORA ADAPTER QUALITY ON EIGENLORA PC INITIALIZATION

To evaluate EigenLoRA's robustness to adapter quality and its resistance to noise, we conducted an ablation study on a subset of tasks of the NLU experiment subsection 4.2. Specifically, we generated EigenLoRA adapters using LoRA matrices with varying levels of random noise added. The results are shown in Table 15

| Noise Level | CoLA | MRPC | RTE | STS-B | Avg |
|---|---|---|---|---|---|
| 5% | 60.51 | 85.45 | 74.73 | 89.9 | 77.65 |
| 15% | 57.53 | 83.09 | 72.92 | 89.9 | 75.86 |
| 30% | 55.23 | 76.47 | 71.84 | 89.8 | 73.34 |

Table 15: EigenLoRA performance on subset of GLUE task using noisy LoRA adapters for initialization

The results show that EigenLoRA exhibits only minor performance changes even as noise levels significantly increase, indicating some robustness to adapter quality. This suggests that EigenLoRA can still perform effectively without high quality adapters. However, there is a limit to this robustness. If the signal-to-noise ratio (SNR) in the initial LoRA matrices becomes extremely low—where the

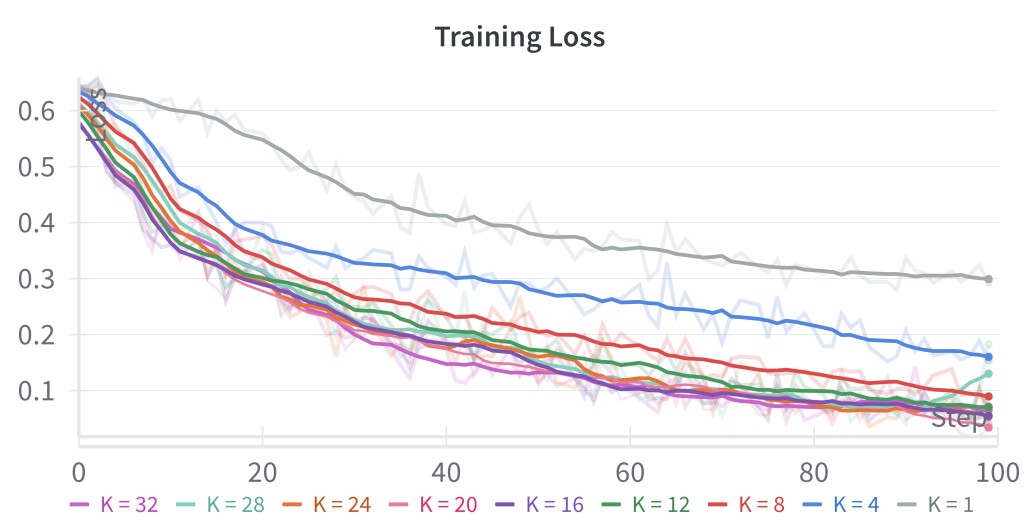

(a) Training Loss Convergence for different numbers of EigenLoRA PCs

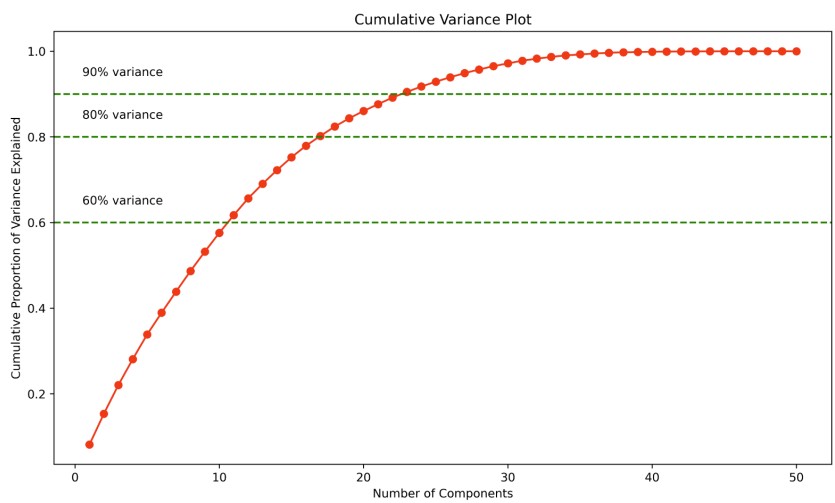

(b) Explained Variance for increasing number of PCs

Figure 11: Ablation of Number of EigenLoRA Principal Components

LoRAs primarily encode noise rather than meaningful information—the effectiveness of EigenLoRA diminishes. In such cases, the principal components (PCs) extracted by EigenLoRA would correspond to random directions in the parameter space. Consequently, EigenLoRA's performance would resemble that of random matrix methods, such as VeRA and NoLA. These methods rely on a large number of random components or bases to approximate meaningful results. While they can achieve reasonable performance, they require fine-tuning a substantially larger number of weights associated with these large number of random components, leading to less efficient learning compared to EigenLoRA. This highlights an important consideration: for EigenLoRA to maintain its efficiency and effectiveness, the initial LoRA matrices must contain at least a minimal level of meaningful signal. This requirement ensures that EigenLoRA can leverage the structured information encoded in the LoRAs while avoiding the inefficiencies of purely random approaches.

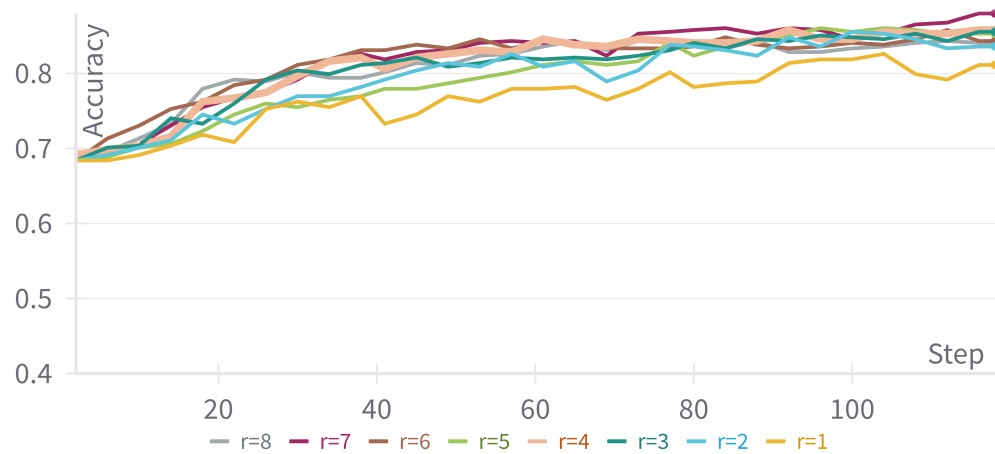

Figure 12: Ablation for the EigenLoRA's $r$ hyperparameter. This experiment was done for the MRPC task in the GLUE benchmark.

## B.4 FORWARD PASS AND BACKWARD PASS FLOPS

While it is obvious that EigenLoRA utilized significantly less number of model parameters as the number of tasks in a domain increase, we show that even in terms of floating point operations on a single task, EigenLoRA is more efficient than LoRA for our experiments. Even for a single task, the number of floating point operations or multiply-accumulate operations in a forward pass for EigenLoRA is lower than LoRA for all our experiments. Here are the comparisons of the floating point operations (FLOPs) for the forward (fwd FLOPs) and including backward pass (fwd+bwd FLOPs) for each of the Image Classification and GLUE benchmark (batch size = 1) (MFLOPs - MegaFlops):

| Method | Training Parameters | fwd FLOPs | fwd+bwd FLOPs |
|---|---|---|---|
| LoRA | 1.2M | 97,930 MFLOPS | 293,800 MFLOPS |
| VeRA | 25K | 106,390 MFLOPS | 319,170 MFLOPS |
| EigenLoRA | 12K | 97,030 MFLOPS | 291,080 MFLOPS |

Table 16: Floating Point Operation calculations for GLUE Benchmark experiment

| Method | Training Parameters | fwd FLOPs | fwd+bwd FLOPs |
|---|---|---|---|
| LoRA | 36K | 33,773.8 MFLOPS | 101,322 MFLOPS |
| VeRA | 18K | 33,744.8 MFLOPS | 101.234 MFLOPS |
| EigenLoRA | 96 | 33,730.2 MFLOPS | 101,191 MFLOPS |

Table 17: Floating Point Operation calculations for Image Classification experiment

## C BROADER IMPACT AND IMPLICATIONS

This work presents a novel parameter-efficient method for deep learning methods utilizing open source, pretrained Low-Rank Adaptation (LoRA) models. By substantially reducing the computational and memory demands of training and inference, our approach creates a more sustainable and environmentally friendly deep learning paradigm. Our method democratizes accessibility to larger models, making them accessible to researchers and practitioners with limited resources. Furthermore, by harnessing pretrained models, our method can accelerate development and diminish the need for extensive data collection. However, we recognize the inherent risks associated with the use of

pretrained models. These include potential biases (racial, gender, etc.), explicit content, since there is no guarantee of the data or method used in training the model, and the potential presence of malicious code. Appropriate caution is advised when using unverified, open-source models.

