# OpenReview forum: "EigenLoRA: Recycle trained Adapters for Resource Efficient Adaptation and Inference"
_ICLR.cc/2025/Conference — Submitted to ICLR 2025_

### Official Review · Reviewer_cTqP · 2024-10-30

**Soundness:** 4
**Presentation:** 4
**Contribution:** 3
**Rating:** 6
**Confidence:** 2

**Summary:**

The authors propose a new method based on LoRA called EigenLoRA. This method focuses on the K principal component of the weight matrices of LoRA. This allows for a reduction in the number of parameters to learn during training, time computation, and inference memory usage. After an introduction to their method, the authors propose several experimental studies to show the benefit of EignenLoRA for different modalities and tasks.

**Strengths:**

- The paper is very well written. It is easy to follow the paper and understand the needs of the field and how they improve LoRA with eigendecomposition.
- The method proposed is an simple but efficient improve of the LoRA method
- A complete experimental part is proposed where EigenLoRA is tested over four different datasets. Each experiment show a benefit of the method.

**Weaknesses:**

Major comment:
- The introduction and related work are short, and it could have been interesting to see a comparison with another method in the experiment study.
- In part 4.2.1, the part with the initialization is not very clear. Why does initialization add a lot of parameters to EigenLoRA? Why does it help for specific tasks (see MRPC), and why is it disruptive for others (see RTE)? Do you have any intuitions?-

Minor comments:
- Tables 2 and 3 have no bold performances, which is harder to read.
- No x-axis title for figure 4

**Questions:**

- In practice, how do you choose the K? Do you always have to find the K best representative principal components?

---

> ### Author Response · Authors · 2024-11-20
> **Response to Reviewer cTqP**
>
> We thank the reviewer for taking out time to review our work.
>
> *W1. The introduction and related work are short, and it could have been interesting to see a comparison with another method in the experiment study.*
>
> Due to space constraints, we had to keep the introduction and related work sections concise, but we have made a deliberate effort to cover the most critical points in these sections. Additionally, we will include further discussion in the Appendix to address any gaps.
>
> In our diverse experiments, we compare our method with PiSSA, VeRA, and LoRA (Section 4), as these are the most broadly applicable across the range of tasks we evaluate, including image classification, natural language understanding (NLU), and text-to-image generation. In contrast, many other methods are often limited to specific domains or setups, such as large language models (LLMs), making them less relevant to our diverse scope.
>
> Moreover, given time and compute constraints, and in line with the resource-efficient motivation of our work, we have prioritized the most impactful comparisons and leave additional baselines for future investigation. This approach ensures that our evaluations remain focused and aligned with the practical efficiency goals of our method. However, if the esteemed reviewer believes that a very relevant work should be added, please let us know and we will add it in our paper. In addition to the experiments in the paper, we include 3 more:
>
> Here are the results for **CommonSense Reasoning and Arithmetic Reasoning** tasks using LLAMA1-7B model.
>
> | **Method**           | **Trainable Param** | **BoolQ** | **PIQA** | **SIQA** | **HellaSwag** | **WinoGrande** | **ARC-e** | **ARC-c** | **OBQA** | **Average** |
> |:--------------------:|:-------------------:|:---------:|:--------:|:--------:|:-------------:|:--------------:|:---------:|:---------:|:--------:|:-----------:|
> | **LoRA (r=32)**      | 56.1M               | 68.53     | 71.22    | 67.91    | 80.91         | 73.64          | 72.13     | 56.74     | 62.2     | 69.16       |
> | **EigenLoRA (k=32)** | 0.3M                | 68.55     | 72.05    | 71.1     | 78.1          | 74.24          | 71.13     | 56.7      | 64.54    | 69.55       |
>
> | **Method**           | **Trainable Param** | **MultiArith** | **GSM8K** | **AddSub** | **AQuA** | **SingleEq** | **SVAMP** | **Average** |
> |:--------------------:|:-------------------:|:--------------:|:---------:|:----------:|:--------:|:------------:|:---------:|:-----------:|
> | **LoRA (r=32)**      | 56.1M               | 95             | 37.5      | 83.3       | 18.9     | 84.4         | 52.11     | 61.86       |
> | **EigenLoRA (k=32)** | 0.3M                | 95.66          | 39.91     | 83.04      | 17.64    | 84.65        | 50.25     | 61.85       |
>
>
> We also conduct **3D object pose estimation** finetuning experiment using a modified Resnet-101. The task of 3D object pose estimation involves the prediction of three rotation parameters (azimuth, elevation, in-plane rotation) of an object relative to the camera. The pose estimation error between the predicted rotation matrix and the ground truth rotation matrix is given as
> $\Delta (R_{pred}, R_{gt}) = \frac{\vert\vert \log b (R_{pred}^{\intercal} R_{gt}) \vert\vert_F}{\sqrt{2}} $
> We show the results for $\frac{\pi}{6}$ threshold for this experiment.
>
> | **Method**          | **Trainable Parameters** | **Airplane** | **Motorbike** | **Boat** | **Bottle** | **Bus** | **Car** | **Average** |
> |:-------------------:|:-------------------:|:---------:|:--------:|:--------:|:----------:|:-------:|:-------:|:-------:|
> | **LoRA (r=16)**     | 215K                | 79.9      | 80.1     | 71.5     | 89.8       | 90.1    | 96.6    | 84.67   |
> | **VeRA (r=256)**    | 40K                 | 68.4      | 72.4     | 64.3     | 88.4       | 87.2    | 94.4    | 79.18   |
> | **EigenLoRA (K=2)** | **16K**                 | 81.4      | 80       | 71.4     | 90         | 92.3    | 97.5    | **85.43**   |
>
> We hope these *7 sets of experiments* are sufficient to show the efficacy of our method.

---

> ### Author Response · Authors · 2024-11-20
> **Response to Reviewer cTqP - Part 2**
>
> *W2.  In part 4.2.1, the part with the initialization is not very clear. Why does initialization add a lot of parameters to EigenLoRA? Why does it help for specific tasks (see MRPC), and why is it disruptive for others (see RTE)? Do you have any intuitions?*
>
> In Table 2, $\text{EigenLoRA}_{init}$  represents a specialized experiment designed to compare our method against finetuning initialization approaches like PiSSA. In this setup, both the EigenLoRA principal components (PCs) and coefficients are trainable, which explains the increased number of parameters.
>
> Our results demonstrate that EigenLoRA also functions as a competitive adapter weight initialization method, delivering better performance and faster convergence than PiSSA. The improved performance observed for some tasks compared to standard EigenLoRA is likely due to the availability of more trainable parameters, enabling deeper convergence. Conversely, the occasional performance drop can be attributed to the complexity and non-convexity of deep network optimization, which may cause the initialized EigenLoRA to deviate from its initial weight subspace, even when the optimal solution resides within it.
>
> ---------------------------------
>
> *Minor Comments*
>
> We have added bold highlights in Table 2 and 3. There is an x-axis in Figure 4 in bottom right corner, however it seems to be largely occluded by the lines. We have therefore updated this figure and moved the x-axis label.
>
> -------------------------
>
> *Q1. In practice, how do you choose the K? Do you always have to find the K best representative principal components?*
>
> We have updated Section B.1 which discusses this. in practice, we find that choosing top $K$ EigenLoRA PCs that correspond to $50-80$% *explained variance* of the LoRAs used to initialize the EigenLoRA is sufficient for robust performance. This can also be done by using a validation dataset and ablating a range of $K$ values.
>
> Since the top $K$ principal components contain most of the information or variance, we do find, theoretically and empirically, that choosing them would give the best performance.

---

> ### Comment · Reviewer_cTqP · 2024-11-26
>
> I thank the authors for their responses. The new experiments reinforce my choice that this paper is clear and brings novelty to the community. I think a score of 6 perfectly reflects the quality of the paper, so I'll keep my score.

---

### Official Review · Reviewer_FWNT · 2024-11-03

**Soundness:** 3
**Presentation:** 2
**Contribution:** 3
**Rating:** 5
**Confidence:** 3

**Summary:**

This paper introduces EigenLoRA, a parameter-efficient fine-tuning method centers on extracting a shared subspace of essential directions (PCs) from multiple already-trained LoRA adapters. Instead of learning a full new adapter for each task, it only learns a small set of task-specific coefficients that combine these principal components to achieve adaptation. This approach dramatically reduces the number of parameters needed for fine-tuning and allows efficient task-switching by loading just the task-specific coefficients rather than full adapter matrices.

**Strengths:**

1. EigenLoRA significantly reduces the number of trainable parameters required for new tasks, which is particularly valuable for low-resource devices or applications with strict memory constraints. By isolating task-specific coefficients and retaining shared principal components, EigenLoRA offers a reduction in memory usage during inference.

2. The method is empirically validated on diverse domains—including image classification, natural language understanding, and text-to-image generation—demonstrating its versatility and robustness across modalities and tasks.

3. EigenLoRA’s initialization in a shared principal subspace results in faster convergence during training, allowing it to reach or exceed baseline performance more efficiently than traditional approaches.

**Weaknesses:**

1. The paper omits evaluations on certain baseline datasets, such as the RESISC45 dataset for image classification, which was included in VeRA’s evaluations.

2. The specific ViT model used for image classification is not clearly identified, and the paper does not fully explain why certain settings yield different results compared to those in baseline studies.

3. Unlike VeRA, which includes comparisons across different model backbones (for image classification), this paper only evaluates a single backbone, limiting the assessment of its generalizability.

4. While baseline methods conduct evaluations on E2E and instruction-tuning tasks, this paper neither includes these benchmarks nor provides a rationale for their exclusion.

5. The proposed method uses customized hyperparameter settings, including varying learning rates and schedulers, while baselines adhere to a fixed set of hyperparameters, potentially compromising the fairness of comparisons.

6. Although the authors mention a study of the effect of the number of principal components K in Section A.2.2, no detailed results are provided.

7. In lines 883-884, the choice of rank r = 8 is not explained, nor is there an exploration of the impact of different r values on performance.

8. Some figures appear to be low-resolution screenshots from wandb, affecting readability of visual results.

**Questions:**

See the weakness part

---

> ### Author Response · Authors · 2024-11-20
> **Response to Reviewer FWNT**
>
> We thank the reviewer for their time.
>
> *W1.The paper omits evaluations on certain baseline datasets, such as the RESISC45 dataset for image classification, which was included in VeRA’s evaluations.*
>
> We have added the results of RESISC45 dataset in the paper. Here are the results:
>
> | **Method**          | **Training Parameters** | **RESISC45** |
> |:-------------------:|:-----------------------:|:------------:|
> | **Base Model**      | -                       | 92.62        |
> | **LoRA (r=4)**      | +147K                   | 95.79        |
> | **VeRA**            | +18K                    | 92.81        |
> | **EigenLoRA (k=2)** | +96                     | 95.40        |
>
> As can be observed, our method achieves performance similar to LoRA using only a fraction of LoRA's trainable parameters.
>
> Further, we provide results for **CommonSense Reasoning and Arithmetic Reasoning** tasks using LLAMA1-7B model.
>
> | **Method**           | **Trainable Param** | **BoolQ** | **PIQA** | **SIQA** | **HellaSwag** | **WinoGrande** | **ARC-e** | **ARC-c** | **OBQA** | **Average** |
> |:--------------------:|:-------------------:|:---------:|:--------:|:--------:|:-------------:|:--------------:|:---------:|:---------:|:--------:|:-----------:|
> | **LoRA (r=32)**      | 56.1M               | 68.53     | 71.22    | 67.91    | 80.91         | 73.64          | 72.13     | 56.74     | 62.2     | 69.16       |
> | **EigenLoRA (k=32)** | 0.3M                | 68.55     | 72.05    | 71.1     | 78.1          | 74.24          | 71.13     | 56.7      | 64.54    | 69.55       |
>
> | **Method**           | **Trainable Param** | **MultiArith** | **GSM8K** | **AddSub** | **AQuA** | **SingleEq** | **SVAMP** | **Average** |
> |:--------------------:|:-------------------:|:--------------:|:---------:|:----------:|:--------:|:------------:|:---------:|:-----------:|
> | **LoRA (r=32)**      | 56.1M               | 95             | 37.5      | 83.3       | 18.9     | 84.4         | 52.11     | 61.86       |
> | **EigenLoRA (k=32)** | 0.3M                | 95.66          | 39.91     | 83.04      | 17.64    | 84.65        | 50.25     | 61.85       |
>
> Further, we also conduct **3D object pose estimation** finetuning experiment using a modified Resnet-101. The task of 3D object pose estimation involves the prediction of three rotation parameters (azimuth, elevation, in-plane rotation) of an object relative to the camera. The pose estimation error between the predicted rotation matrix and the ground truth rotation matrix is given as
> $\Delta (R_{pred}, R_{gt}) = \frac{\vert\vert \log b (R_{pred}^{\intercal} R_{gt}) \vert\vert_F}{\sqrt{2}} $
> We show the results for $\frac{\pi}{6}$ threshold for this experiment.
>
> | **Method**          | **Trainable Parameters** | **Airplane** | **Motorbike** | **Boat** | **Bottle** | **Bus** | **Car** | **Average** |
> |:-------------------:|:-------------------:|:---------:|:--------:|:--------:|:----------:|:-------:|:-------:|:-------:|
> | **LoRA (r=16)**     | 215K                | 79.9      | 80.1     | 71.5     | 89.8       | 90.1    | 96.6    | 84.67   |
> | **VeRA (r=256)**    | 40K                 | 68.4      | 72.4     | 64.3     | 88.4       | 87.2    | 94.4    | 79.18   |
> | **EigenLoRA (K=2)** | **16K**                 | 81.4      | 80       | 71.4     | 90         | 92.3    | 97.5    | **85.43**   |
>
> We will be adding both of these experiments to our paper with additional details.
>
> We would like to clarify to the esteemed reviewer that while we include VeRA as a comparative method, our work does not replicate or claim to follow their choice of experiments. We kindly ask the reviewer to evaluate our work on its own merits.
>
> Our experiments cover a significantly more diverse range of evaluations—spanning Image Classification, 3D Pose Estimation, Natural Language Understanding, Commonsense Reasoning, LoRAHub, Arithmetic Reasoning, and Text-to-Image Generation. These tasks utilize models such as ViT-base, ResNet101, Roberta-base, FlanT5, LLama1-7B, and Stable Diffusion. In contrast, VeRA evaluates only $4$ tasks, primarily centered on Natural Language Processing. Furthermore, as we found out, VeRA’s current code cannot be easily adapted for tasks like text-to-image generation using Stable Diffusion models, unlike our approach.
>
> Our experimental choices are driven by the goal of demonstrating the practical applicability and benefits of our method across a wide range of domains, tasks, and modalities while adhering to our compute and space limits. We believe this breadth underscores the robustness and versatility of our work compared to VeRA.

---

> > ### Author Response · Authors · 2024-11-20
> > **Response to Reviewer FWNT - Part 2**
> >
> > *W2. The specific ViT model used for image classification is not clearly identified, and the paper does not fully explain why certain settings yield different results compared to those in baseline studies.*
> >
> > The specific model used in our Image classification is ViT-base model, specifically, vit-base-patch16-224. We have added this information in the paper.
> >
> > We appreciate the reviewer’s feedback but are uncertain which specific experiment is being referred to. This ambiguity is understandable, as we re-ran all baseline experiments rather than directly copying results reported in prior works. For our implementations of LoRA, VeRA, and PiSSA, we relied on the HuggingFace _PeFT_ and _Diffusers_ libraries, systematically tuning hyperparameters (e.g., batch size) within our computational constraints. We have presented the best results achieved through this process, which, while not identical, closely align with the original reported performances.
> >
> > For certain experiments, such as Commonsense Reasoning, discrepancies between reported and achieved results have been observed and documented by other researchers (e.g., [LLM-Adapters GitHub Issue #38](https://github.com/AGI-Edgerunners/LLM-Adapters/issues/38)). Additionally, we optimized both LoRA and VeRA to reduce the number of trainable parameters while maintaining comparable performance, ensuring a fairer comparison with EigenLoRA.
> >
> > If the reviewer’s concern pertains to Image Classification experiments, we clarify that our experimental setup is not the same as VeRA’s (as mentioned above) (refer to Section 4.1). Instead, it aligns more closely with approaches commonly used in Continual or Federated Learning contexts, highlighting our focus on demonstrating practical utility across diverse scenarios.
> >
> > ------------------------
> > *W3. Unlike VeRA, which includes comparisons across different model backbones (for image classification), this paper only evaluates a single backbone, limiting the assessment of its generalizability.*
> >
> > As previously mentioned, we do not claim to follow VeRA's experimental choices. For Image Classification, VeRA conducts experiments on two backbones—ViT-B and ViT-L. In contrast, we choose a single backbone, ViT-base, as we find it sufficient for achieving robust results, especially considering our focus on compute efficiency. We do not believe that using one backbone instead of two for this experiment limits the _generalizability_ of our method.
> >
> > In terms of overall generalization, our work now presents results from 7 distinct experiments, each using a completely different backbone (not just variants of the same architecture) across 7 diverse tasks. These span image data, 3D data, natural language, and multimodal image-text modalities. By comparison, VeRA offers results from only 4 experiments, limited to image classification and natural language processing tasks.
> >
> > Given the breadth and diversity of our experimental evaluation across multiple domains, we respectfully believe that the reviewer's concerns regarding generalizability are not justified, particularly when compared to VeRA.
> >
> > -------

---

> ### Author Response · Authors · 2024-11-21
> **Response to Reviewer FWNT - Part 3**
>
> *W4. While baseline methods conduct evaluations on E2E and instruction-tuning tasks, this paper neither includes these benchmarks nor provides a rationale for their exclusion.*
>
> Thank you for raising this point. Please refer to W1 for our results on Commonsense Reasoning and Arithmetic Reasoning, alongside 3D Pose Estimation.
> Importantly, we would like to point out that both E2E and instruction-tuning are only trained on a single dataset and since EigenLoRA requires some initial LoRA adapters to begin with, these benchmarks are trivially out-of-scope in their current setup. We are working on establishing a framework by identifying useful LoRAs that can be used in each of these tasks. Further, there is no publicly available code for VeRA for the E2E benchmark, and LoRA's custom implementation for this task is not easily adaptable to the HuggingFace _PeFT_ library that we use for consistency. Nonetheless, we are actively working to establish a unified experimental setup for all methods and will strive to include the E2E benchmark in our draft as soon as possible. Adding this benchmark will increase the total number of experiment setups in our paper to 8, covering more domains than LoRA and VeRA papers combined.
>
> As discussed in previous sections, we do not claim to follow VeRA’s experimental choices. Our rationale for selecting experiments is to demonstrate EigenLoRA's versatility and applicability across a wide range of domains, modalities, and tasks. These choices are also influenced by practical constraints, including limited space and access to compute resources.
>
> When comparing our experiments with those of VeRA and LoRA, it becomes evident that VeRA lacks coverage in several key areas. Specifically, VeRA does not include experiments on Text-to-Image generation using Stable Diffusion models, LoRAHub, 3D Pose Estimation (or any 3D computer vision-related task, which is notably distinct from image classification), Commonsense Reasoning, or Arithmetic Reasoning. A similar observation can be made regarding LoRA paper, which also does not address such diverse domains.
>
> In summary, we believe that our experimental setup already surpasses VeRA and LoRA in terms of diversity and domain coverage, reflecting the broader applicability of EigenLoRA. We appreciate the reviewer’s patience as we continue to enhance our benchmarks, and respectfully urge the reviewer to reevaluate this point, as our current experiments already demonstrate significantly broader applicability and diversity compared to VeRA and LoRA, highlighting the robustness and versatility of EigenLoRA across domains.
>
> --------
> *W5. The proposed method uses customized hyperparameter settings, including varying learning rates and schedulers, while baselines adhere to a fixed set of hyperparameters, potentially compromising the fairness of comparisons.*
>
> This appears to be a misunderstanding, and we have added further clarification on the same in the Appendix A1.
>
> For VeRA, LoRA and PiSSA, we experimented with a range of learning rates, from higher to lower, along with three different scheduling approaches: ReduceLRonPlateau, Linear, and Cosine, and other hyperparameters, within our compute constraints. The hyperparameters that yielded the best average performance were selected for further experimentation.
>
> On the contrary, we did not perform a thorough investigation of all hyperparameters for EigenLoRA due to compute and time limitations. This was also because we observed that often the initially selected hyperparameters, for example, a high learning rate paired with ReduceLRonPlateau or Linear scheduler, demonstrated satisfactory performance, thereby conserving computational resources. For comparison, we redid the GLUE experiment for EigenLoRA using exactly the hyperparameter of LoRA (which have been optimized for LoRA's best performance) and observe similar results.
>
> |                               | CoLA  | MRPC  | QNLI  | RTE   | SST-2 | STS-B | Avg   |
> |-------------------------------|-------|-------|-------|-------|-------|-------|-------|
> | EigenLoRA (ReduceLRonPlateau) | 59.81 | 87    | 92.73 | 77.62 | 94.15 | 90.58 | 83.65 |
> | EigenLoRA (Linear)            | 61.58 | 89.71 | 92.21 | 75.33 | 93.35 | 89.56 | 83.63 |
>
> If the reviewer prefers, we can replace the results of our comparative methods in the paper with the suboptimal experimental results obtained using the hyperparameter settings aligned with EigenLoRA.
>
> ---------
> *W6. Although the authors mention a study of the effect of the number of principal components K in Section A.2.2, no detailed results are provided.*
>
> We ablate the choice of number of EigenLoRA principal components in Section B.1. Concisely, in practice, we find that choosing top $K$ EigenLoRA PCs that correspond to $50-80$% *explained variance* of the LoRAs used to initialize the EigenLoRA is sufficient for robust performance. This can also be done by using a validation dataset and ablating a range of $K$ values.

---

> > ### Author Response · Authors · 2024-11-22
> > **Response to Reviewer FWNT - Part 4**
> >
> > *W7. In lines 883-884, the choice of rank r = 8 is not explained, nor is there an exploration of the impact of different r values on performance.*
> >
> > As mentioned in the same line, $r$ hyperparameter in EigenLoRA does not equate to the $rank$ parameter found in LoRA and its variants. It is simply the dimensionality of the trainable coefficients and even $r=1$ works fairly well. Increasing the value tends to improve performance slightly. We have added the ablation for this parameters in section B.1.
> >
> > ----------
> >
> > *W8. Some figures appear to be low-resolution screenshots from wandb, affecting readability of visual results.*
> >
> > We apologize for any perceived low-resolution figures in the draft. While none of our main figures are screenshots, we have updated all WandB graphs in the paper for improved clarity. Given the large number of figures in our submission—some of which, such as the qualitative results for text-to-image generation, require particularly high visual quality—we had to downsample certain Appendix images to comply with the ICLR file size limits imposed by OpenReview. We will continue to refine the balance between figure quality and file size restrictions and kindly request that the reviewer considers these limitations, which we don't have control over, when evaluating our submission.

---

> ### Author Response · Authors · 2024-11-25
> **Author Response**
>
> Thank you for the follow up. We discuss this in Reviewer HC8T Q1.
>
> EigenLoRA relies on pre-trained LoRA adapters to initialize its principal components, but this does not significantly limit its applicability in most real-world scenarios. For widely used models, there is already a rich ecosystem of publicly available LoRA adapters. For instance, the HuggingFace library currently hosts **1000+ LoRA adapters for the LLAMA-2-7B model and over 4000 adapters for Stable-Diffusion-XL-base-1.0**, many of which are underutilized. EigenLoRA provides a practical and efficient way to recycle these adapters, compress them into a compact form, and extract meaningful features for future tasks, drastically reducing fine-tuning costs and improving resource utilization.
>
> *In cases where no pretrained adapters exist—such as for novel architectures or entirely new benchmarks—we do not claim that EigenLoRA is trivially applicable*. In such situations, the initial LoRA training process is necessary to create adapters that can then be used to initialize EigenLoRA. This one-time investment in LoRA training is subsequently offset by the significant efficiency gains EigenLoRA offers for future tasks. For fair benchmarking, the trainable parameters for both the initial LoRA training and EigenLoRA should be accounted for, as this reflects the true resource costs while showcasing the long-term benefits of adopting EigenLoRA for downstream adaptation and inference.
>
> In future, we also plan to research if any near-domain EigenLoRAs would be applicable in these new domains or benchmarks. However, right now, our method is focused on tasks and models with publicly available adapters, which are available for vast majority of common tasks and models, which underlines EigenLoRA's utility in practical scenarios.
>
> Thus, EigenLoRA is positioned as a framework that maximizes efficiency by leveraging existing adapters wherever they are available. In domains or benchmarks without adapters, the LoRA training process becomes a prerequisite, but this step is well justified by the downstream gains in efficiency and scalability. By offering a flexible and resource-aware solution, EigenLoRA addresses both adapter-rich and adapter-scarce environments, enabling practical and cost-effective model adaptation without overclaiming its direct applicability to every possible scenario.

---

### Official Review · Reviewer_mfg3 · 2024-11-03

**Soundness:** 3
**Presentation:** 2
**Contribution:** 2
**Rating:** 3
**Confidence:** 4

**Summary:**

This paper introduces EigenLoRA, a parameter-efficient fine-tuning method that improves upon LoRA by recycling information from previously trained adapters through identifying a principal subspace shared by adapters trained on related tasks within a domain, allowing new tasks to be adapted by learning coefficients for pre-extracted principal components rather than full LoRA matrices, which results in using up to 100x fewer parameters and achieving up to 2x faster convergence during training while improving memory efficiency by ~18x when switching between multiple tasks during inference by only loading lightweight coefficients rather than full adapter matrices, demonstrating wide applicability across different modalities and domains with both theoretical foundations including approximation bounds for reconstruction error and practical validation, positioning it as a resource-efficient solution particularly suitable for edge devices and personalization applications.

**Strengths:**

This work presents comprehensive experiments and evaluations on language and CV models, image generation, and achieves good results.

**Weaknesses:**

1. Absent evidence of practical cost optimization: In fact, the reduced number of training parameters but still requires a large number of fixed parameters in the forward process. These fixed parameters also incur significant additional overhead. That is, this approach does not significantly reduce the time and memory needed in fine-tuning. Just reducing training parameters is not practical and diminishes fine-tuning performance.


2. Lack of novelty: Actually, this work just utilizes SVD as the initialization of LoRA and dynamically selects the fine-tuning of the eigens, which  already many studies in previous works. Adaptive Budget Allocation for Parameter-Efficient Fine-Tuning (ICLR), DyLoRA: Parameter-Efficient Tuning of Pre-trained Models using Dynamic Search-Free Low-Rank Adaptation. EACL.


3. Lack of solid evaluations: Evaluation In Vision Transformer, the GLUE benchmark heavily relies on hyperparameter tuning, and the image generation fine-tuning effect is not sensitive to the design of the LoRA. In other words, those presented experiments  are not a solid reflection of the performance of the proposed method. I suggest that the authors present more evidence of in Commonsense Reasoning and instruction fine-tuning tasks in recent LLMs. e.g., Llama 3 8B and 70B, Llama 3.2 1B and 3B, DeepSeekMoE，Mixtral-8x7B.

4. Numbers Trainable Parameters such as +0, in Tables 1 and 3 are confusing. I believe that +0 does not mean that there is no additional fine-tuning overhead . The authors need to use other cost metrics.

5. Absence of theoretical Support: why partial updated LoRA variants like this one would be better than full parameter updated LoRA variants.

6. Lack of recent parameter-efficient LoRA method discussion. e.g., LoRA-XS: Low-Rank Adaptation with Extremely Small Number of Parameters, NoRA: Nested Low-Rank Adaptation for Efficient Fine-Tuning Large Models.

**Questions:**

See weaknesses.

---

> ### Author Response · Authors · 2024-11-19
> **Response to Reviewer mfg3**
>
> Respected Reviewer, Thank you for taking the time to review our work.
>
> *W1. Absent evidence of practical cost optimization*
>
> We believe the reviewer has misunderstood the amount of fixed parameters that is required by EigenLoRA in the forward pass. **EigenLoRA does not require a large number of fixed number of parameters**.
>
> Even for a single task, the number of floating point operations or multiply-accumulate operations in a forward pass for EigenLoRA is lower than LoRA for all our experiments. Here are the comparisons of the floating point operations (FLOPs) for the forward (fwd FLOPs) and including backward pass (fwd+bwd FLOPs) for each of the Image Classification and GLUE benchmark (batch size = 1) (MFLOPs - MegaFlops):
>
> *GLUE Benchmark*
> | **Method**    | **Params** | **fwd FLOPs**  | **fwd+bwd FLOPs** |
> |:-------------:|:----------:|:--------------:|:-----------------:|
> | **LoRA**      | 1.2M       | 97,930 MFLOPS  | 293,800 MFLOPS    |
> | **VeRA**      | 25K        | 106,390 MFLOPS | 319,170 MFLOPS    |
> | **EigenLoRA** | **12K**        | **97,030 MFLOPS**  | **291,080 MFLOPS**    |
>
> *Image Classification*
> | **Method**    | **Params** | **fwd FLOPs**   | **fwd+bwd FLOPs** |
> |:-------------:|:----------:|:---------------:|:-----------------:|
> | **LoRA**      | 36K        | 33,773.8 MFLOPS | 101,322 MFLOPS    |
> | **VeRA**      | 18K        | 33,744.8 MFLOPS | 101.234 MFLOPS    |
> | **EigenLoRA** | **96**         | **33,730.2 MFLOPS** | **101,191 MFLOPS**    |
>
> As can be seen, even for a single forward pass on a single task, EigenLoRA uses less fixed parameters (and therefore less FLOPs), and therefore **no additional overhead** relative to LoRA, which contradicts the esteemed reviewer's claim regarding significant additional overhead.
>
> This saving becomes significant in real world practical scenarios when the number of tasks and therefore, LoRAs, gets larger and we need to save and switch a large number of adapter parameters for new tasks. As shown in Section 4.3, we can use a single set of EigenLoRA PCs i.e. a single set of fixed parameters for a number of tasks, leading to **18x reduction in memory requirement**. For large model serving systems[1] which are used to serve a large number of users and required to optimize LoRA switching and loading, EigenLoRA provides a practical cost optimized solution, which can lead to faster response times, lower operational costs and lower memory footprint, since then EigenLoRA PCs can be fixed for a large number of tasks as shown in Section 4. Additionally, we show that the significantly reducing the number of trainable parameters leads to faster optimization (see Section 4.2.1 and Figure 4)  (up to 1.5x faster than LoRA).
>
> Given all these practical benefits, we find that the reviewer's assertion of 'absent evidence' for practical benefits is not substantiated by the presented findings.
>
> [1] Kwon et al. - Efficient Memory Management for Large Language Model Serving with PagedAttention
>
> --------------
> *W2. Lack of novelty: Actually, this work just utilizes SVD as the initialization of LoRA and dynamically selects the fine-tuning of the eigens, which already many studies in previous works. Adaptive Budget Allocation for Parameter-Efficient Fine-Tuning (ICLR), DyLoRA: Parameter-Efficient Tuning of Pre-trained Models using Dynamic Search-Free Low-Rank Adaptation. EACL.*
>
> The esteemed reviewer appears to have misunderstood the core contribution of our work. Our novelty does not lie in the truncated SVD initialization of LoRA. Instead, we demonstrate that pretrained low-rank adapters can uncover a robust principal subspace, which not only enhances further task-specific learning efficiency but also amplifies the benefits of reduced training and inference costs, and other benefits as discussed in previous paragraph and Section 1. To the best of our knowledge, the idea of a learning a shared principal subspace by recycling information from pretrained low rank adapters for guiding further efficient finetuning has not been explored before, which underlines the novelty of our work. If we are mistaken, please cite these papers in your comment and we'd be happy to compare EigenLoRA with any such method.
>
> Further, reviewer cites AdaLoRA and DyLoRA as comparative methods. We’d like to note that both of these methods try to optimize the rank allocation in LoRA. AdaLoRA adjust the rank by repeated SVD and importance scoring of the low rank adapter, while DyLoRA does so by truncating the A and B matrices (there is no SVD involved). There is no concept of a principal subspace in both of these methods and they are finetuned from random initialization.
> Additionally, we do not try to solve the problem of dynamic rank-allocation in our work as EigenLoRA does not have a parameter which is equivalent to LoRA “rank”. Therefore, although great works, we don’t find AdaLoRA and DyLoRA to be connected to our method.

---

> ### Author Response · Authors · 2024-11-20
> **Response to Reviewer mfg3 - continued**
>
> *W3. Lack of solid evaluations: Evaluation In Vision Transformer, the GLUE benchmark heavily relies on hyperparameter tuning, and the image generation fine-tuning effect is not sensitive to the design of the LoRA. In other words, those presented experiments are not a solid reflection of the performance of the proposed method. I suggest that the authors present more evidence of in Commonsense Reasoning and instruction fine-tuning tasks ..*
>
> We thank the reviewers for their experimental suggestions. In this work, we do not do explore extensive hyperparameter tuning for EigenLoRA given our focus on efficiency and constraints on available compute resources. It is possible that better performance may be obtained for EigenLoRA with hyperparameter tuning for most of our experiments.
>
> It'll also be great if the esteemed reviewers can clarify what they mean by "image generation fine-tuning effect is not sensitive to the design of the LoRA" and add relevant citations, so we can do a thorough analysis for our method. In our experiments, however, we came across major challenges in the text-to-image generation experiments as the vast majority of PeFT methods do not code  implementations which can be easily applied to these generative models, We also found that methods like VeRA did not trivially work well in this complex task setup.
>
> Here are the results for **CommonSense Reasoning and Arithmetic Reasoning** tasks using LLAMA1-7B model. Please note that we reran the LoRA baseline using the LLM-adapters[1] codebase and could not achieve the reported performance (although we get close). This problem is documented in the repository issues (https://github.com/AGI-Edgerunners/LLM-Adapters/issues/38).
>
> | **Method**           | **Trainable Param** | **BoolQ** | **PIQA** | **SIQA** | **HellaSwag** | **WinoGrande** | **ARC-e** | **ARC-c** | **OBQA** | **Average** |
> |:--------------------:|:-------------------:|:---------:|:--------:|:--------:|:-------------:|:--------------:|:---------:|:---------:|:--------:|:-----------:|
> | **LoRA (r=32)**      | 56.1M               | 68.53     | 71.22    | 67.91    | 80.91         | 73.64          | 72.13     | 56.74     | 62.2     | 69.16       |
> | **EigenLoRA (k=32)** | 0.3M                | 68.55     | 72.05    | 71.1     | 78.1          | 74.24          | 71.13     | 56.7      | 64.54    | 69.55       |
>
> | **Method**           | **Trainable Param** | **MultiArith** | **GSM8K** | **AddSub** | **AQuA** | **SingleEq** | **SVAMP** | **Average** |
> |:--------------------:|:-------------------:|:--------------:|:---------:|:----------:|:--------:|:------------:|:---------:|:-----------:|
> | **LoRA (r=32)**      | 56.1M               | 95             | 37.5      | 83.3       | 18.9     | 84.4         | 52.11     | 61.86       |
> | **EigenLoRA (k=32)** | 0.3M                | 95.66          | 39.91     | 83.04      | 17.64    | 84.65        | 50.25     | 61.85       |
>
>
> Further, we also conduct **3D object pose estimation** finetuning experiment using a modified Resnet-101. The task of 3D object pose estimation involves the prediction of three rotation parameters (azimuth, elevation, in-plane rotation) of an object relative to the camera. The pose estimation error between the predicted rotation matrix and the ground truth rotation matrix is given as
> $\Delta (R_{pred}, R_{gt}) = \frac{\vert\vert \log b (R_{pred}^{\intercal} R_{gt}) \vert\vert_F}{\sqrt{2}} $
> We show the results for $\frac{\pi}{6}$ threshold for this experiment.
>
> | **Method**          | **Trainable Parameters** | **Airplane** | **Motorbike** | **Boat** | **Bottle** | **Bus** | **Car** | **Average** |
> |:-------------------:|:-------------------:|:---------:|:--------:|:--------:|:----------:|:-------:|:-------:|:-------:|
> | **LoRA (r=16)**     | 215K                | 79.9      | 80.1     | 71.5     | 89.8       | 90.1    | 96.6    | 84.67   |
> | **VeRA (r=256)**    | 40K                 | 68.4      | 72.4     | 64.3     | 88.4       | 87.2    | 94.4    | 79.18   |
> | **EigenLoRA (K=2)** | **16K**                 | 81.4      | 80       | 71.4     | 90         | 92.3    | 97.5    | **85.43**   |
>
> We will be adding both of these experiments to our paper with additional details.
>
> Including the above experiments, we believe that our work offers the most diverse set of evaluations—spanning *Image Classification, 3D Pose Estimation, Natural Language Understanding, Commonsense Reasoning, Arithmetic Reasoning, and Text-to-Image Generation*—compared to all the methods referenced by the esteemed reviewer. This breadth of experimentation provides a comprehensive and robust validation of our approach, despite time and compute constraints.
>
> [1] Hu et al. - LLM-Adapters: An Adapter Family for Parameter-Efficient Fine-Tuning of Large Language Models.
>
> [2] Wang et al. - NeMo: Neural Mesh Models of Contrastive Features for Robust 3D Pose Estimation.

---

> ### Author Response · Authors · 2024-11-20
> **Response to Reviewer mfg3 - continued (2)**
>
> *W4. Numbers Trainable Parameters such as $+0$, in Tables 1 and 3 are confusing. I believe that $+0$ does not mean that there is no additional fine-tuning overhead . The authors need to use other cost metrics.*
>
> $+0$ reflects zero-shot performance over the base model, for which the performance is also given. An initialized EigenLoRA with frozen PCs and randomly initialized coefficients is used for this evaluation. There is no finetuning done nor are any adapter parameters optimized for this experiment using backpropagation. Please suggest a cost metric that the reviewer finds applicable for assessing the results of these experiments and we will add them to our paper.
>
> -----------------------------------------------
>
> *W5. Absence of theoretical Support: why partial updated LoRA variants like this one would be better than full parameter updated LoRA variants.*
>
> We provide initial theoretical support and intuition for EigenLoRA in Section 3.1, where we explain why finding principal subspaces may give good approximations to the original weights with fewer parameters. If the optimal weight lies on or close to the underlying subspace that can be found by our method, this would imply that our method can outperform basic LoRA for which this weight region may remain unexplored or unknown due to the learning dynamics and random initialization. Please note this intuition only applies to EigenLoRA and not to other LoRA variants that the reviewer may be referring to. By convention, we could create a theoretical toy model which would align perfectly with our theoretical construction and show perfect performance, however, we believe our diverse set of experiments in Section 4 provides a stronger evidence for our method in real world scenarios. In fact, our theoretical construction shows that it is possible to have high approximation error in estimating new weights through principal subspaces if the new task has orthogonal components, but in practice we rarely find that to be the case. We are also able to explain failure cases (figure 7) with our theoretical construction (as increasing the number of EigenLoRA PCs decreases the chances of failure).
>
> We would like to note that none of our comparative methods or those cited by the reviewer (LoRA, VeRA, PiSSA, AdaLoRA, DyLoRA, NoRA, LoRA-XS) provide any theoretical proofs, whereas our work does include one. While we acknowledge that further theoretical investigations could be valuable, we believe this can be explored in future work. We kindly ask the reviewers to reevaluate their comment regarding the need for additional theoretical results in light of this context.
>
> ---------------------------------------
>
> *W6. Lack of recent parameter-efficient LoRA method discussion. e.g., LoRA-XS: Low-Rank Adaptation with Extremely Small Number of Parameters, NoRA: Nested Low-Rank Adaptation for Efficient Fine-Tuning Large Models.*
>
> We thank the reviewers for bringing these two methods to our attention. While we have made a genuine effort to include all relevant papers, the rapid pace of advancements in this field and the sheer volume of research make it challenging to capture every work. We appreciate the reviewers' input and have will be citing these works in our paper and will continue to update our related work discussion with more works.
>
> We note that both the LoRA-XS and NoRA papers have not yet been published at a peer-reviewed venue. According to ICLR guidelines, we are not required to compare our work to such papers. Furthermore, the current arXiv versions of LoRA-XS and NoRA were uploaded after July 1, 2024. As per the ICLR policy, “if a paper was published (i.e., at a peer-reviewed venue) on or after July 1, 2024, authors are not required to compare their own work to that paper.” Additionally, it appears that both submissions are currently under review at ICLR, creating a unique situation that may raise ethical concerns.
> In light of these guidelines, we respectfully request the reviewer to reevaluate our method regarding this perceived weakness.
>
> During first implementation, we observe that LoRA-XS requires a significant amount of time (30 minutes to 5 hours on an A5000 GPU, depending on the rank, e.g., 2 to 64) to initialize the A and B matrices for LLAMA-7B models, whereas EigenLoRA’s initialization time is negligible. Additionally, LoRA-XS incurs a higher overhead in terms of FLOPs. Moreover, LoRA-XS lacks experiments in the image domain or with vision-language models, unlike our work, leaving its applicability in these areas untested.
>
> While we couldn’t analyze NoRA due to the absence of publicly available code, we acknowledge that its experiments appear more comprehensive than those of LoRA-XS. However, we believe EigenLoRA offers greater diversity in its experimental evaluation.

---

### Official Review · Reviewer_HC8T · 2024-11-05

**Soundness:** 2
**Presentation:** 3
**Contribution:** 3
**Rating:** 6
**Confidence:** 4

**Summary:**

The paper introduces EigenLoRA, an approach for parameter-efficient fine-tuning that leverages the principal subspaces of low-rank adapters (LoRA) trained on various domain-specific tasks. EigenLoRA reduces the parameter count by up to 100x and optimizes memory efficiency, benefiting tasks such as inference on resource-constrained devices. By identifying a domain’s principal subspace, EigenLoRA offers a lightweight way to perform new task adaptations by only learning coefficients within this subspace, avoiding full weight reinitialization. The approach is validated on image classification and NLP benchmarks, demonstrating competitive performance with significantly lower computational overhead.

**Strengths:**

1. EigenLoRA achieves high parameter and memory efficiency, cutting parameters by up to 100x. This makes it ideal for low-resource deployments.
2. The method performs well across various tasks, matching LoRA’s results with far fewer parameters, and proving its versatility.
3. Figures and tables are well-designed.
4. EigenLoRA’s low memory demand fits well with real-world edge applications, like personal devices with limited resources.

**Weaknesses:**

1. EigenLoRA’s success depends on high-quality adapters, which might be limiting in under-researched domains.
2. Guidance on picking the right number of principal components would help with applying this method across diverse tasks.
3. Failure cases need more examples to help users understand when and why the method might struggle.
4. While suited for edge devices, more real-world benchmarks would strengthen claims about efficiency in low-memory environments to meet ICLR standards.

**Questions:**

Q1. How do you envision managing the dependency on trained adapters in a practical deployment setting? Are there scenarios where this reliance could hinder flexibility?
Q2. Can EigenLoRA be extended or modified to handle tasks involving multi-modal or cross-domain data? If so, what challenges do you foresee?

---

> ### Author Response · Authors · 2024-11-23
> **Response to Reviewer HC8T**
>
> Respected Reviewer, Thank you for taking the time to review our work.
> *W1. EigenLoRA’s success depends on high-quality adapters, which might be limiting in under-researched domains.*
>
> To evaluate EigenLoRA’s robustness to adapter quality and its resistance to noise, we conducted an ablation study on a subset of tasks of the NLU experiment. Specifically, we generated EigenLoRA adapters using LoRA matrices with varying levels of random noise added.
>
> | Noise Level | CoLA  | MRPC  | RTE   | STS-B | Avg   |
> |-------------|-------|-------|-------|-------|-------|
> | 5%      | 60.51 | 85.45 | 74.73 | 89.9  | 77.65 |
> | 15%      | 57.53 | 83.09 | 72.92 | 89.9  | 75.86 |
> | 30%      | 55.23 | 76.47 | 71.84 | 89.8  | 73.34 |
>
> The results show that EigenLoRA exhibits only minor performance changes even as noise levels significantly increase, indicating some robustness to adapter quality. This suggests that EigenLoRA can still perform effectively without high quality adapters. However, there is a limit to this robustness. If the signal-to-noise ratio (SNR) in the initial LoRA matrices becomes extremely low—where the LoRAs primarily encode noise rather than meaningful information—the effectiveness of EigenLoRA diminishes.
> In such cases, the principal components (PCs) extracted by EigenLoRA would correspond to random directions in the parameter space. Consequently, EigenLoRA’s performance would resemble that of random matrix methods, such as VeRA and NoLA. These methods rely on a large number of random components or bases to approximate meaningful results. While they can achieve reasonable performance, they require fine-tuning a substantially larger number of weights associated with these large number of random components, leading to less efficient learning compared to EigenLoRA.
> This highlights an important consideration: for EigenLoRA to maintain its efficiency and effectiveness, the initial LoRA matrices must contain at least a minimal level of meaningful signal. This requirement ensures that EigenLoRA can leverage the structured information encoded in the LoRAs while avoiding the inefficiencies of purely random approaches.
>
> -----------
> *W2. Guidance on picking the right number of principal components would help with applying this method across diverse tasks.*
>
> Please refer to Section B1 for a discussion on the same.
>
> ----------
> *W3. Failure cases need more examples to help users understand when and why the method might struggle.*
>
> We’ve added more discussion on failure cases in Section B2. In summary, Figure 7 highlights a failure case of EigenLoRA caused by selecting too few principal components (PCs), excluding important task-specific information. This issue can be resolved by increasing the number of PCs. Another potential failure scenario, though not observed in our experiments, could occur if the domain gap between the low-rank adapters and the downstream task is too large, which might be mitigated by making only a subset of PCs trainable. Additionally, we found that biases in the LoRAs used to initialize EigenLoRA can persist in complex tasks like Text-to-Image generation although this problem of model bias is not exclusive to EigenLoRA and is applicable to other gradient descent optimized deep models.

---

> ### Author Response · Authors · 2024-11-23
> **Response to Reviewer HC8T - Part 2**
>
> *W4. While suited for edge devices, more real-world benchmarks would strengthen claims about efficiency in low-memory environments to meet ICLR standards.*
>
> Thank you for the suggestion. In addition to the experiments provided in the draft, we conducted three additional benchmarks—Commonsense Reasoning, Arithmetic Reasoning, and 3D Object Pose Estimation. These tasks also have significant real-world applicability across various domains in addition to the experiments present in our draft,
>
> Commonsense Reasoning: Demonstrates the ability of EigenLoRA to handle general knowledge-based reasoning tasks effectively.
> Arithmetic Reasoning: Highlights robustness in mathematical reasoning tasks, which are relevant for applications like financial analysis or data modeling on edge devices.
> 3D Object Pose Estimation: Validates performance in spatial reasoning tasks, which have direct implications for robotics, augmented reality, and autonomous systems.
>
> Here are the results for **CommonSense Reasoning and Arithmetic Reasoning** tasks using LLAMA1-7B model.
>
> | **Method**           | **Trainable Param** | **BoolQ** | **PIQA** | **SIQA** | **HellaSwag** | **WinoGrande** | **ARC-e** | **ARC-c** | **OBQA** | **Average** |
> |:--------------------:|:-------------------:|:---------:|:--------:|:--------:|:-------------:|:--------------:|:---------:|:---------:|:--------:|:-----------:|
> | **LoRA (r=32)**      | 56.1M               | 68.53     | 71.22    | 67.91    | 80.91         | 73.64          | 72.13     | 56.74     | 62.2     | 69.16       |
> | **EigenLoRA (k=32)** | 0.3M                | 68.55     | 72.05    | 71.1     | 78.1          | 74.24          | 71.13     | 56.7      | 64.54    | 69.55       |
>
> | **Method**           | **Trainable Param** | **MultiArith** | **GSM8K** | **AddSub** | **AQuA** | **SingleEq** | **SVAMP** | **Average** |
> |:--------------------:|:-------------------:|:--------------:|:---------:|:----------:|:--------:|:------------:|:---------:|:-----------:|
> | **LoRA (r=32)**      | 56.1M               | 95             | 37.5      | 83.3       | 18.9     | 84.4         | 52.11     | 61.86       |
> | **EigenLoRA (k=32)** | 0.3M                | 95.66          | 39.91     | 83.04      | 17.64    | 84.65        | 50.25     | 61.85       |
>
>
> Further, we also conduct **3D object pose estimation** finetuning experiment using a modified Resnet-101. The task of 3D object pose estimation involves the prediction of three rotation parameters (azimuth, elevation, in-plane rotation) of an object relative to the camera. The pose estimation error between the predicted rotation matrix and the ground truth rotation matrix is given as
> $\Delta (R_{pred}, R_{gt}) = \frac{\vert\vert \log b (R_{pred}^{\intercal} R_{gt}) \vert\vert_F}{\sqrt{2}} $
> We show the results for $\frac{\pi}{6}$ threshold for this experiment.
>
> | **Method**          | **Trainable Parameters** | **Airplane** | **Motorbike** | **Boat** | **Bottle** | **Bus** | **Car** | **Average** |
> |:-------------------:|:-------------------:|:---------:|:--------:|:--------:|:----------:|:-------:|:-------:|:-------:|
> | **LoRA (r=16)**     | 215K                | 79.9      | 80.1     | 71.5     | 89.8       | 90.1    | 96.6    | 84.67   |
> | **VeRA (r=256)**    | 40K                 | 68.4      | 72.4     | 64.3     | 88.4       | 87.2    | 94.4    | 79.18   |
> | **EigenLoRA (K=2)** | **16K**                 | 81.4      | 80       | 71.4     | 90         | 92.3    | 97.5    | **85.43**   |
>
> In addition to the performance benchmarks, we emphasize the memory efficiency and computational benefits of EigenLoRA, which are particularly important for resource-constrained environments like edge devices. As detailed in Section 4.3, EigenLoRA achieves an **18x** memory saving for text-to-image generation models, a result that extends to other tasks. Beyond memory savings in trainable parameters, EigenLoRA typically requires fewer FLOPs than LoRA for both forward and backward passes, leading to additional energy and VRAM savings.
>
> (continued)

---

> ### Author Response · Authors · 2024-11-23
> **Response to Reviewer HC8T - Part 3 (continued)**
>
> *W4. continued*
>
> Even for a single task, the number of floating point operations or multiply-accumulate operations in a forward pass for EigenLoRA is lower than LoRA for all our experiments. Here are the comparisons of the floating point operations (FLOPs) for the forward (fwd FLOPs) and including backward pass (fwd+bwd FLOPs) for each of the Image Classification and GLUE benchmark (batch size = 1) (MFLOPs - MegaFlops):
>
> *GLUE Benchmark*
> | **Method**    | **Params** | **fwd FLOPs**  | **fwd+bwd FLOPs** |
> |:-------------:|:----------:|:--------------:|:-----------------:|
> | **LoRA**      | 1.2M       | 97,930 MFLOPS  | 293,800 MFLOPS    |
> | **VeRA**      | 25K        | 106,390 MFLOPS | 319,170 MFLOPS    |
> | **EigenLoRA** | **12K**        | **97,030 MFLOPS**  | **291,080 MFLOPS**    |
>
> *Image Classification*
> | **Method**    | **Params** | **fwd FLOPs**   | **fwd+bwd FLOPs** |
> |:-------------:|:----------:|:---------------:|:-----------------:|
> | **LoRA**      | 36K        | 33,773.8 MFLOPS | 101,322 MFLOPS    |
> | **VeRA**      | 18K        | 33,744.8 MFLOPS | 101.234 MFLOPS    |
> | **EigenLoRA** | **96**         | **33,730.2 MFLOPS** | **101,191 MFLOPS**    |
>
> As can be seen, even for a single forward pass on a single task, EigenLoRA uses less fixed parameters (and therefore less FLOPs), and therefore **no additional overhead** relative to LoRA.
>
> For edge devices, these benefits translate into:
>
> - Smaller model size: Reducing the storage and memory footprint.
> - Improved training efficiency: Faster training cycles with fewer computational demands.
> - Lower compute requirements: Enabling deployment in low-power environments, such as IoT devices or smartphones.
>
> We are happy to consider running additional benchmarks recommended by the reviewer, within the constraints of our time and computational resources, provided they align with the scope of EigenLoRA. However, we believe our current diverse and extensive set of experiments—spanning tasks with real-world relevance and quantifiable efficiency metrics—already meets the standards of ICLR.
>
> -----------
> *Q1. How do you envision managing the dependency on trained adapters in a practical deployment setting? Are there scenarios where this reliance could hinder flexibility?*
>
> Thank you for the question. EigenLoRA principal components are only initialized by recycling available adapters, meaning there is no direct dependency on trained adapters during practical deployment. Instead, EigenLoRA reuses adapters at the initialization stage, allowing for efficient compression and knowledge transfer without requiring them to remain part of the deployed system.
> For most common tasks, the availability of pretrained LoRA adapters is generally not a limiting factor. For example, as of now, the HuggingFace community library provides **1040 LoRA adapters for the LLAMA-2-7B model and 4000+ adapters for Stable-Diffusion-XL-base-1.0**. Many of these adapters are underutilized, being downloaded and applied only a few times after creation. EigenLoRA offers a practical way to recycle these underused adapters, compress them into a smaller, efficient form, and extract meaningful information for future tasks. This approach significantly reduces fine-tuning costs and maximizes the utility of existing resources.
> In scenarios where absolutely no pretrained adapters are available—such as in under-researched domains or novel architectures—we recommend starting with LoRA for the initial tasks. These LoRA-trained adapters can then be used to create EigenLoRA, enabling more efficient training and inference for subsequent tasks. This bootstrapping approach ensures flexibility while maintaining compatibility with EigenLoRA’s design philosophy.
> Overall, EigenLoRA mitigates reliance on existing adapters by offering a framework that leverages their availability where possible while remaining adaptable in their absence, thereby balancing efficiency and flexibility in practical deployment.
>
> ---
> *Q2. Can EigenLoRA be extended or modified to handle tasks involving multi-modal or cross-domain data? If so, what challenges do you foresee?*
>
> EigenLoRA already handles multimodal data effectively, as shown by its performance in text-to-image generation experiments. Extending EigenLoRA to cross-domain tasks, where it is trained on one domain and applied to another, poses significant challenges due to domain differences, model architecture variations, and disparities in input-output data formats. For example, models optimized for text may not generalize well to tasks involving images or audio without additional adaptations. Addressing these challenges would require advancements in areas such as domain-specific fine-tuning, unified representation learning, or adapter transfer techniques that map EigenLoRA components across architectures. While EigenLoRA’s inherent efficiency and adaptability make it a promising candidate for such tasks, this remains an open problem and a valuable direction for future research.

---

### Meta-Review · Area_Chair_M6vK · 2024-12-17

**Metareview:**

a) This paper introduces a parameter-efficient fine-tuning (PEFT) method that improves upon LoRA by learning a principal subspace of adapters pre-trained on related tasks within a domain. This allows a new task to be adapted faster, with fewer parameters and memory.

b) This method achieves high parameter and memory efficiency, cutting parameters by up to 100x, making it ideal for low-resources devices. The paper presents comprehensive experiments and evaluations on language, computer vision and image generation models, achieving promising results.

c) In contrast to other PEFT methods, this method requires a set of pre-trained adapters to learn the principal subspace. This limits its applicability to only domains where pre-trained adapters are available and makes the comparison with the other approaches (which do not require that) not fully fair. The techniques used are quite standard, so, apart form the application, there is no much technical novelty.

d) The most important reason to reject the paper is the lack of a fair comparative. Authors claim high reduction in terms of computation, memory and parameters, assuming the pre-trained adapters needed for their training as given. While this can be realist in many use-cases, still the comparative is not fully fair. The proposed model requires less resources because part of the training is amortized in the pre-trained PEFT adapters. Instead of putting forward the impressive numbers, authors should have presented in a fairer comparative evaluation advantages and disadvantages of the proposed approach.

**Additional Comments On Reviewer Discussion:**

Rev. HC8T provided a meaningful review highlighting advantages and drawbacks of the proposed approach, but he did not engage in further discussion nor with the authors nor with the other reviewers.

Rev. mfg3 is the most critical on the paper. While I share some points (e.g. reduced novelty in the technique proposed), other claims where not well supported by justifications or references (e.g. lack of solid evaluations) and did not engage with the compelling answers form the authors.

Rev. FWNT provided a fair evaluation, stressing the additional requirements of the proposed method (to have pre-trained adapters) and engaged with the authors to better understand several points. The authors pushed for an increase of score but the rev. explained why they wanted to keep their original score.

Rev. cTqP provided a shallow review due to their limited knowledge in the field. In the final discussion she/he explained that the given score was mostly due to the clarity of the presentation, but he was willing to follow the opinion of the other reviewers.

Overall, I share the evaluation provided by FWNT, while considering also some of the positive and negative points presented by the other reviewers. I think the authors should focus on a fairer presentation and evaluation of their proposed method for a new submission.

---

### Decision · Program_Chairs · 2025-01-22

Reject